# Action-specialized expert ensemble trading system with extended discrete action space using deep reinforcement learning

JoonBum Leem[1], Ha Young Kim[2]*

1 Department of Financial Engineering, Ajou University, Yeongtong-gu, Suwon, Republic of Korea,
2 Graduate School of Information, Yonsei University, Seodaemun-gu, Seoul, Republic of Korea

* haimkgetup@gmail.com

**Data Availability Statement:** The data are held in GitHub, a public repository. https://github.com/JoonBumLeem/Time-series-dataset.

## Abstract

Despite active research on trading systems based on reinforcement learning, the development and performance of research methods require improvements. This study proposes a new action-specialized expert ensemble method consisting of action-specialized expert models designed specifically for each reinforcement learning action: buy, hold, and sell. Models are constructed by examining and defining different reward values that correlate with each action under specific conditions, and investment behavior is reflected with each expert model. To verify the performance of this technique, profits of the proposed system are compared to those of single trading and common ensemble systems. To verify robustness and account for the extension of discrete action space, we compared and analyzed changes in profits of the three actions to our model's results. Furthermore, we checked for sensitivity with three different reward functions: profit, Sharpe ratio, and Sortino ratio. All experiments were conducted with S&P500, Hang Seng Index, and Eurostoxx50 data. The model was 39.1% and 21.6% more efficient than single and common ensemble models, respectively. Considering the extended discrete action space, the 3-action space was extended to 11- and 21-action spaces, and the cumulative returns increased by 427.2% and 856.7%, respectively. Results on reward functions indicated that our models are well trained; results of the Sharpe and Sortino ratios were better than the implementation of profit only, as in the single-model cases. The Sortino ratio was slightly better than the Sharpe ratio.

## Introduction

Recently, trading systems based on machine learning have been actively studied in all fields including the financial field [1–7]. With sufficient data, a machine can efficiently learn patterns, exhibiting the notable advantage of the ability to learn unknown patterns [8]. This feature can be exploited for trading systems and consequently, is actively studied using machine learning in the financial field. Through machine learning, vast amounts of data can be quickly calculated, while an objective judgment of the database can help determine important financial

**Funding:** This research was supported by a grant (19CTAP-C152020-01) from the Technology Advancement Research Program (TARP) funded by the Ministry of Land, Infrastructure and Transport of the Korean government. The funder played no role in the study design, data collection, and analysis, decision to publish, or preparation of the manuscript. The author JB was involved in the study prior to belonging to Daishin Asset Management company, and no funds have been contributed by the company. The specific roles of these authors are articulated in the 'author contributions' section.

**Competing interests:** The authors have declared that no competing interests exist. Author JB was involved in the study prior to belonging to Daishin Asset Management company. There are no commercial sources of funding for this study. There are no patents, products in development or marketed products to declare. This does not alter our adherence to PLOS ONE policies on sharing data and materials.

transactions. Machine learning is largely divided into supervised, unsupervised, and reinforcement learning [9]. In the financial field, the supervised learning method extracts important features from labeled data and uses a classification and prediction model. Many studies have been conducted, ranging from those based on statistical learning theory to those using state-of-the-art machine learning algorithms such as Support Vector Machine (SVM), Random Forest (RF), and Deep Neural Network (DNN) [10–13]. The unsupervised learning method, which uses unlabeled data, is mainly used for clustering and finding patterns in the data using dimension-reduction machine learning techniques such as auto-encoder. One of the representative studies is the Deep Portfolio Theory [14]. In reinforcement learning (RL), which is mainly used in trading systems research, a model-free method that relies on the input of market conditions as a state is applied using a reward function. A representative study by Moody and Saffell [15] that led to numerous subsequent studies examined the optimal portfolio, asset allocation, and trading system using Recurrent Reinforcement Learning (RRL).

Although trading systems research based on RL is actively conducted, there are many challenges, such as difficulties in analyzing and training, which arise from insufficient data or excessive noise [4]. Additionally, RL itself is difficult to train. To improve performance, the ensemble method is one of the most widely used machine learning methods [16]. However, because applying the ensemble method to RL is more difficult than the general machine learning algorithm, it is yet to be applied in automated financial trading systems research. Therefore, we posit that if an ensemble technique specialized for RL is applied to a trading system, the performance of the trading system will improve.

We propose an action-specialized expert ensemble trading system—a novel ensemble method designed specifically for RL—that can reflect investment propensity. This ensemble system consists of action-specialized expert models, with each model specialized for each action examined in the RL for trading systems by using different reward values under specific conditions. Actions of trading systems typically include buying, holding, and selling; we designed an expert single model corresponding to each action to reflect real investment behavior [2, 7, 15]. To create an expert single model, reward values for expert action are controlled. In the common ensemble method, the single model is trained in the same data set with different models or in different data sets with the same model. In other words, various distribution effects for an ensemble can be obtained using these methods. Unlike the common ensemble method, this study employs a method to create an action-specialized expert ensemble model that is specifically developed for buying, holding, and selling actions; we then combine these action-specialized expert models in an ensemble. Our proposed ensemble method is expected to improve performance and reflect characteristics of RL for trading systems. We used soft voting with the softmax function, which is more effective than hard voting, as an ensemble method [16].

To verify our proposed method and check its robustness, we include more action spaces by discretizing, which determines the number of multiple shares of a stock to buy or sell by itself. Previous studies have either only studied the three actions or proceeded to a continuous action space [1–7, 15]. It is well known that as the output of the model network increases, learning becomes more difficult [17]. In a previous study, however, discretizing action spaces yielded a better performance than applying continuous action spaces [18]. Thus, in this study, we extend the number of actions from 3 to 11 and 21, and the quantity of actions is increased by 5 and 10, respectively. Moreover, we expect the network to be able to recognize market risk and control the quantity by itself. One of the purposes of our research is to create a more profitable automated trading system that allows for more investment when data-driven patterns are clearer, such as real investors investing more boldly as compared to the information they receive from the market. Therefore, our model is designed to learn various patterns from data and vary actions to increase profit according to the magnitude of the reward value we designed.

Compared to the existing 3-action models, existing models could not represent the diversity of actions depending on the reward value of the model trained from the data. For example, we could not determine whether the buy signal in the 3-action model is strong or weak. In contrast, our proposed system with more discrete actions is significantly more profitable than the 3-action system because it can buy or sell more, depending on the market situation. More specifically, the trading model with 21 actions can ideally increase profits by up to 10 times, since it can trade more quantities (up to 10 times) for stronger signals than weaker ones.

If the extended action space model can capture the level of obvious patterns from the dynamic market data, it can decide the quantities of investment by itself—depending on the captured level of information. As we give the adaptive signal to our model through controlling reward values by extending action space, we expect that our model can analyze more detailed market information, which includes the degree of both direction and magnitude of market movements. If the proposed model is confident in the market condition, it will invest more in the market. Whereas, if the model is less confident in the market condition, it will adjust the quantity to take a relatively small risk in order to achieve a small loss or a small profit. In this regard, we have produced many experimental results that can support this. Many RL-based trading system studies surveyed have 3-action spaces, and our research is meaningful as it is the pioneering study to attempts this. As expected, our results indicate that Deep Reinforcement Learning (DRL) can learn not only three actions, but also various other actions, depending on the strength of the network signals.

Further, we used three types of reward functions: profit, and the Sharpe and Sortino ratios, to examine the sensitivity of our proposed ensemble model. Generally, profit is a frequently used reward function in RL for trading systems research [2, 7, 15]. Since the Sharpe and Sortino ratios are calculated using profits and volatilities, they are suitable reward functions to train networks for RL [3, 4]. Thus, to compare the performance of reward functions, we consider not only profit, but also volatility.

Our experiment employed three extensively used data sets—S&P500, Hang Seng Index (HSI), and Eurostoxx50—that efficiently exhibit different price movements for the period from January 1987 to December 2017 [2, 6, 7, 15]. For the same period, we divided these data sets into training and test periods of 20 and 11 years, respectively. Our basic model is based on the Deep Q-Network (DQN), and we employ online learning on the test data set. While the DQN is well known for combining the Q-learning algorithm with DNN [19], online learning is a method in which data become available in a sequential order, especially in a test data set [20].

The remainder of this paper is organized as follows. Section 2 describes the related research. Section 3 discusses the related methodologies of DQN, reward functions, and our proposed method. Section 4 analyzes our data sets in various ways. Section 5 describes the experiments of our proposed model and explains our methodology. Section 6 reveals experimental results and conducts detailed analyses. Section 7 concludes and suggests future applications.

## Related work

In the financial field, there are many recent studies that employ machine learning for forecasting, classification, dimension reduction, and trading. In this section, we describe the relevant literature.

### Supervised learning in the financial field

Most financial studies using supervised learning attempted to predict price fluctuations or trends. Trafalis and Ince [10] predicted stock prices using SVM based on statistical learning theory and compared it to Radial Basis Function (RBF). Huang, Nakamori, and Wang [21] also performed Nikkei225 index prediction using SVM, and compared Linear Discriminant

Analysis (LDA), Quadratic Discriminant Analysis (QDA), and Elman Backpropagation Neural Networks for performance evaluation. Tsai and Wang [22] studied the stock price prediction model by combining Artificial Neural Network (ANN) and a decision tree to improve the performance of a single model. Patel, Shah, Thakkar, and Kotecha [11] conducted two studies. First, they used four forecasting models—ANN, SVM, RF, and Naïve Bayes—to predict stock market index prices and trends. Second, they proposed a prediction model combining ANN, RF, and Support Vector Regression (SVR) and indicated better performance than the single model. Recent research has proposed a model to forecast a financial market crisis using DNN, the boosting method [23], and ModAugNet, which adds two modules to prevent overfitting, and improves prediction performance [13].

## Unsupervised learning in the financial field

Most financial research that employed unsupervised learning were conducted in the direction of dimension reduction using an auto-encoder. As a representative study, a deep portfolio theory composed of 4 steps—auto-encoder, calibrating, validating, and verifying—was developed by Heaton, Polson, and Witte [14]. Chong, Han, and Park [24] compared reconstruction error, stock price fluctuation, and prediction using Principal Component Analysis (PCA), auto-encoder, and Restricted Boltzmann Machine (RBM). Bao, Yue, and Rao [25] conducted price prediction using the model combining Wavelet Transform, Stacked Auto-Encoder (SAE), and Long-Short Term Memory (LSTM). First, wavelet transformation is applied to the time series data to remove noise, while high-level features of data are extracted by SAE. Subsequently, the processed and transformed data are used for stock price prediction using LSTM.

## Reinforcement learning in the financial field

Finance-related research using RL has been conducted mainly to improve the performance of trading algorithms. Moody and Saffell [15] conducted a study on the optimal portfolio, asset allocation, and trading system using RRL, which became the basis for a significant amount of research. They compared various methods using profit, Differential Sharpe Ratio (DSR), and Downside Deviation Ratio (DDR) as reward functions and indicated that RRL is better than Q-learning. Since then, many researchers have used RRL and DSR: Almahdi and Yang [3] proposed an optimal variable weight portfolio allocation model using RRL and Expected Maximum Drawdown (EMD) to solve the dynamic asset allocation problem; Deng, Bao, Kong, Ren, and Dai [4] improved performance by combining the RRL with the fuzzy DNN model, which analyzes the market; Huang [5] used small replay memory, added feedback signal, and sampled long sequences to improve the existing research with Deep Recurrent Q-Network (DRQN). Wang et al. [2] studied the trading system using DQN and compared it to the RRL strategy performance. This study became the basis of our research to compare and improve performance. In addition, another study added three ideas of the trading system by applying the existing DQN, which technically added to and changed the network. First, the number of stocks traded using DNN is determined. Second, the decision is suspended by analyzing the confusing situation. Lastly, this study uses transfer learning to account for the lack of data [7]. We summarize trading system studies using RL in Table 1 and compare our experiments with those of other papers.

## Ensemble methods in the financial field

There is an ensemble learning method for machine learning that performs much better than existing single models. Tsai, Lin, Yen, and Chen [26] proposed a model combining majority voting and bagging that indicated better performance than a single model or existing ensemble

**Table 1. Summary of trading system studies using reinforcement learning.**

| Authors (year) | State | Action | Reward | Data description (Training:Test) | Method |
|---|---|---|---|---|---|
| Saud Almahdi, Steve Y. Yang (2017) | 104 (weekly 2 years) | 3 (-1, 0, 1) | SR STR CR | 5 Funds Jan. 2011–Dec. 2015 (6:4) | RRL |
| Yang Wang et al. (2017) | 200 (daily delta price) | 3 (-1, 0, 1) | Long-term return (100 days) | 2 Index Jan. 2001–Dec. 2015 (4:11) Online-learning | DQN |
| Gyeeun Jeong, Ha Young Kim (2019) | 200 (daily close price) | 3 (-1, 0, 1) + 1 (# of shares) | Long-term return (200 days) | 4 Index Jan. 1987–Dec. 2017 Jan. 2008–Dec. 2017 Apr. 1991–Dec. 2017 Jul. 1997–Dec. 2017 (approx. 4:1:5) Online-learning | DQN + Extra networks |
| John Moody, Matthew Saffell (2001) | 84 (monthly price) | 3 (-1, 0, 1) | Profit DSR DDR | S&P500, T-Bill Jan. 1950–Dec. 1994 (4:5) | RRL |
| Huang, Chien-Yi (2018) | 198 (Time 3, Market 12x16, Position 3) | 3 (-1, 0, 1) | Log return | 12 Currency Jan. 2012–Dec. 2017 (tick) Online-learning | DRQN |
| Yue Deng et al. (2017) | 150 (minute price, 50x3) | 3 (-1, 0, 1) | SR TP | IF future, silver, sugar Jan. 2014–Sep. 2015 (1:5) Jan. 2014–Jan. 2015 (2:5) Online-learning | FDRNN + DRL |
| Parag C. Pendharkar et al. (2018) | 4 (yearly asset statement) | 5 (0:10, 2.5:7.5, 5:5, 7.5:2.5, 10:0) | Profit DSR DDR | S&P500, T-note, AGG 1976–2016 (26:15) | SARSA Q-learning |
| Authors of the present study | 200 (daily close price) | 3 (-1, 0, 1) 11 (±5~ ±1, 0) 21 (±10~±1, 0) | Long-term return (100days) Sharpe ratio Sortino ratio | 3 Index Jan. 1987–Dec. 2017 (21:10) Online-learning | DQN + Expert Ensemble |

SR: Sharpe ratio, STR: Sterling ratio, CR: Calmar ratio, DSR: differential Sharpe ratio, DDR: downside deviation ratio, TP: total profit, RRL: recurrent reinforcement learning, DQN: deep Q-network, DRQN: deep recurrent Q-network, FDRNN: fuzzy deep recurrent neural network, DRL: deep reinforcement learning, SARSA: on-policy reinforcement learning algorithm

methods. Booth, Gerding, and McGroarty [27] proposed a trading system based on weighted ensembles of RF that is specialized in seasonality effects and improves profitability and prediction accuracy. Giacomel, Galante, and Pereira [28] proposed an ensemble network that approached stock price forecasting as a rising or falling classification problem and simulated it by applying it to North American and Brazilian stock markets. Yang, Gong, and Yang [29] proposed a DNN ensemble model that predicts the Shanghai index and the Shenzhen Stock Exchange index using a bagging method. Weng, Lu, Wang, Megahed, and Martinez [30] proposed a model that predicts short-term stock prices using four ensemble methods: neural network regression ensemble, SVR ensemble, boost regression ensemble, and RF regression.

## Continuous and discrete action space in reinforcement learning

There are also studies on continuous and discrete action space in RL. However, a majority of these are related to games or robotics, with only a few studies related to finance. In general, the action space of RL in most environments is continuous; therefore, it is inappropriate to apply a discrete action space [7, 31, 32]. This is reflected in research by Google Deep Mind [33] and the OpenAI team [34]. However, in some real case studies, discretizing action spaces has been shown to be more effective than applying continuous action spaces [35]. Therefore, discretization of actions can be said to improve performance. According to recent studies by OpenAI, this may be because a discrete probability distribution is more expressive than a multivariate Gaussian or because discretization of actions makes the learning of a favorable advantage function potentially easier [18]. Based on this evidence, we believe that extending the discrete action space in this study could be a more efficient approach for the asset allocation problem than what can be accessed as a continuous action space. In addition, we can extend this study to solve the asset allocation problem that exists in the continuous action space with transfer learning by first learning it as a discrete action space problem.

## Methodology

### DQN (Deep Q-network)

Unlike supervised and unsupervised learning from the static environment data, RL is a methodology wherein an agent directly explores environment data, confirms correlated rewards, and establishes policies for optimal action. The objective of RL is to find an optimal policy that maximizes the expected sum of discounted future rewards [36]. These rewards of optimal policy start with choosing the optimal value for each action, which is called the optimal Q-value. RL generally solves problems that can be defined in the Markov Decision Process (MDP). The elements of RL are represented by $(S,A,P,R,\gamma)$, where $S$ is a finite set of states, $A$ is a finite set of actions, $P$ is a state transition probability matrix, $R$ is a reward function, and $\gamma$ is a discount factor. The process of RL is described in Fig 1—the agent observes the state $s_t$ from the environment at time $t$ and selects action $a_t$ [19]. Subsequently, as a result of this action, we receive reward $r_t$ from the environment and obtain the next environment $s_{t+1}$, which is changed by the action $a_t$. If the reward is determined by both the state and action, we can define the action value function $Q_\pi(s_t,a_t)$ as follows:

$$Q_\pi(s_t, a_t) = \mathbb{E}_\pi\left[\sum_{i=0}^\infty \gamma^i r_{t+i+1} | s_t, a_t\right] \tag{1}$$

From this action value function, we can represent the optimal action-value function $Q^*(s_t, a_t)$, maximizing the future reward amount as indicated in Eq (1). The optimal action $a^*(s_t)$ can be obtained from Eq (3) as follows.

$$Q^*(s_t, a_t) = \max_\pi Q_\pi(s_t, a_t) \tag{2}$$

$$a^*(s_t) = \underset{a_t}{\operatorname{argmax}} Q^*(s_t, a_t) \tag{3}$$

Finally, the optimal action value function can be represented by the Bellman equation in Eq (4) [37].

$$Q^*(s_t, a_t) = \mathbb{E}[r_{t+1} + \gamma \max_{a_{t+1}} Q(s_{t+1}, a_{t+1}) | s_t = s, a_t = a] \tag{4}$$

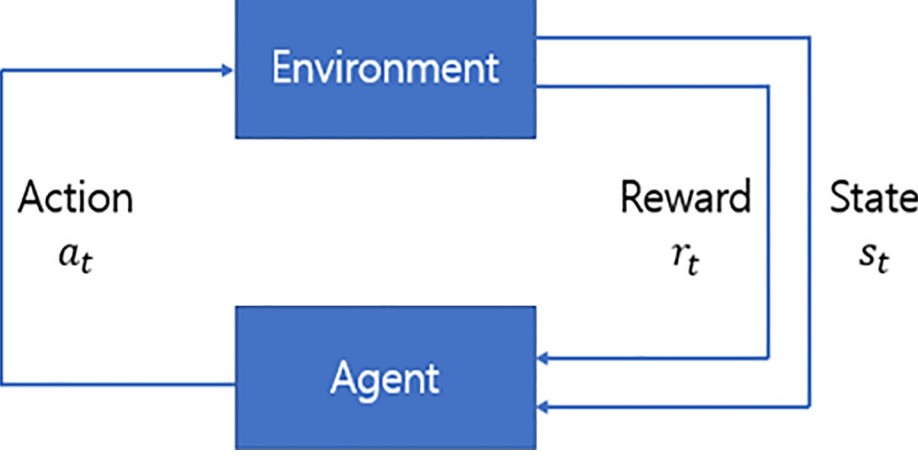

**Fig 1. Interaction of agent and environment in reinforcement learning.**

The basic idea of the RL algorithm is to estimate the action value function by repeatedly calculating and updating the aforementioned Bellman equation. If the iteration is infinite, the action value function converges, and the result is the optimal action value function. The Q-network is composed of a neural network to find the Q-value. The DQN is a deeply structured network that learns by minimizing the loss function, shown by Eq (5).

$$loss_t(w_t) = \mathbb{E}[(y_t - Q(s_t, a_t; w_t))^2] \tag{5}$$

$$y_t = \max_{a_{t+1}} [r_t + \gamma Q(s_{t+1}, a_{t+1}; w_{t-1})|s_t, a_t] \tag{6}$$

where $w_t$ is the weight of the network. This is a neural network that approximates the Q-function, and it is trained in the supervised approach. Hence, it needs a label, and RL uses the target Q-value as the label, i.e., $y_t$ in Eq (6). To obtain the target Q-value, we need the target Q-network that fixes the weight every few steps. Thus, the DQN will train to minimize the loss until the next few steps, repeating this process until convergence. In this study, we attempt to construct a trading system in three index environments by the DQN method and establish an extended discrete action space and action-specialized expert ensemble method.

## Reward functions

The reward function is a guide for model-free RL. The network of RL updates through the value of the reward function. We describe our reward functions as follows.

## Profit function

The most common reward function is the profit function, which has been used in previous studies [2]. This function is outlined as follows:

$$r_{t\_profit} = \left(1 + a_t \times \frac{p_t - p_{t-1}}{p_{t-1}}\right) \frac{p_{t-1}}{p_{t-n}} \tag{7}$$

where $r_{t\_profit}$ is the profit reward function at time $t$, $a_t$ is the action selected at time $t$ by the agent, and $p_t$ is the closing price at time $t$. Eq (7) is an appropriate function for RL because it represents long-term returns over $n$ periods and is less volatile than daily returns. This equation consists of one-day and long-term gross returns. Therefore, $r_{t\_profit}$ is the same as $1 + r_n = (1 + r)\frac{p_{t-1}}{p_{t-n}}$ when $a_t = 1$. In this equation, we assume that the action for sale is -1, the action for hold is 0, and the action for purchase is 1. In the experiment, however, we assume that the action for sale is 0, the action for hold is 1, and the action for purchase is 2 because the network will output only positive numbers. Using long-term returns is useful because they can be considered as long-term stock trading or investments.

## Sharpe and Sortino ratios

Another representative reward function is the Sharpe ratio [38], which can reflect profit and volatility. The Sortino ratio is similar to the Sharpe ratio, and their equations are as follows:

$$r_{t\_sharpe} = \frac{Average(\sum_{i=1}^{I} R_i)}{Standard\,Deviation(\sum_{i=1}^{I} R_i)} \tag{8}$$

$$r_{t\_sortino} = \frac{Average(\sum_{i=1}^{I} R_i)}{Standard\,Deviation_{below}(\sum_{i=1}^{I} R_i)} \tag{9}$$

where $r_{t\_sharpe}$ is the Sharpe ratio reward function at time $t$, $R_i$ is the daily return with the number of multiple shares of a stock by action $a_t$, and $Average(\sum_{i=1}^{I} R_i)$ is the average return over period $I$. $StandardDeviation(\sum_{i=1}^{I} R_i)$ is the standard deviation of daily returns over period $I$. $I$ is the window size for calculating average and standard deviation of returns. $r_{t\_sortino}$ is the Sortino ratio reward function at time $t$, and $StandardDeviation_{below}(\sum_{i=1}^{I} R_i)$ is the standard deviation of the daily return below zero for $I$ period. The Sortino ratio only considers the aspect of volatility of the loss because the volatility under profit conditions is not important. In modern portfolio theory, high Sharpe and Sortino ratios indicate high profit without a large fluctuation. Due to these characteristics of the two ratios, they are appropriate reward functions of RL. We assume that the risk-free rate is 0 (i.e., r_f = 0), and this assumption makes the Sharpe and Sortino ratios invariant to leverage; hence, the leverage effect remains the same regardless of the amount of investment. For instance, consider that there are two profits: 5% and 3%. The average is 4% and the standard deviation is about 1.4%. Therefore, the Sharpe ratio is approximately 2.83. For the leverage effect that is to expand investing, if the number of multiple shares of a stock is five, then profits are 25% and 15%, respectively. Its average is 20%, and the standard deviation is approximately 7.1%. In the end, the Sharpe ratio is approximately 2.83, same as before, indicating that the assumption makes the Sharpe ratio invariant to leverage.

## Proposed single model—action-specialized expert model

The action-specialized expert models are created by adjusting the reward function values under specific conditions. The concept of our proposed single model is to develop an expert model of each action that reflects investors' behavior. For instance, if someone is inclined to buy to generate profit, then we can reflect this behavior tendency in an expert model specialized for an aggressive investor. As aforementioned, we can create various action-specialized expert models with investment strategies that are effective for analyzing buying, selling, and holding actions. In other words, the expert model for buying yields a larger reward value when profit is high, and the model works well in increasing price periods. Similarly, the expert model for selling yields a larger reward value when absolute profit is high, and the model performs well in the dropping period. Further, the expert model for holding yields a large reward when the holding is in the range of profit from -0.3 to 0.3%.

The reward function of the expert model is expressed by the following Eq (10).

$$
r_t^{expert} = \begin{cases} r_t \times m, & \text{if } a_t \text{ is an expertaction in the range of profit} \\ & \text{according to Table 4 below} \\ r_t, & \text{otherwise} \end{cases} \tag{10}
$$

where $r_t$ is the reward value, $m$ is the predetermined positive constant and $m{\geq}1$. For the $m$ (predetermined positive constant), we constructed the range of profit based on profit distribution, and divided it into buy, hold, and sell actions based on the threshold. We set the threshold at 0.3% because it is used as a general transaction cost, and it is possible to prevent a loss by choosing a holding strategy if it does not generate more than 0.3% profit. In Eq (10), $m$ is applied step-by-step—depending on the importance of profit and frequency. As frequency varies according to the profit interval, the absolute value of profit is important. Table 2 indicates the design of predetermined positive constants of the expert model by profit interval. Due to this conditional reward function $r_t^{expert}$, we can control the reward, and through this equation, we used the adjusted reward value to develop the proposed single expert model.

By controlling the reward value with $m$, we can create the enhanced model for specific action according to the reward value. In detail, we modify the reward function by multiplying

**Table 2. Predetermined positive constants of the expert model by profit interval.**

| Expert model: Buy | | Expert model: Hold | | Expert model: Sell | |
|---|---|---|---|---|---|
| Range of profit | Predetermined positive constant (*m*) | Range of profit | Predetermined positive constant (*m*) | Range of profit | Predetermined positive constant (*m*) |
| -∞– 0.3% | 1 | -∞–-0.3% | 1 | -∞–-5% | 10 |
| 0.3–1% | 3 | -0.3–0.3% | 7 | -5 –-3% | 7 |
| 1–2% | 5 | 0.3 –∞% | 1 | -3 –-2% | 6 |
| 2–3% | 6 | | | -2 –-1% | 5 |
| 3–5% | 7 | | | -1 –-0.3% | 3 |
| 5 –∞% | 10 | | | -0.3 –∞% | 1 |

it and *m* for learning the action-specialized expert model when the model makes a correct decision. For example, according to Table 2, the buy-specialized expert model obtains the enhanced reward value that is *m* times larger than the common reward value when its decision is correct in the range of profit. The enhanced compensation is only applied when the decision is correct. If the decision is wrong, the reward value is small or under zero but not at the enhanced penalty value. In other words, the model obtains larger reward values when it works well in the specific action, and so becomes the specific action-specialized expert model. Thus, each action-specialized expert model of buy, hold, and sell can be created by controlling the enhanced reward function with *m*. In addition, we apply the extended discrete action space and it makes the reward value larger than the 3-action space. The extended action space helps the model determine whether the action is strong or weak. Specifically, the buy action in the 3-action model is only one, whereas, the buy actions in the 11-action model are five—which means buying 1 to 5 shares. The action of buying 1 share is similar to a weak buy action whereas the action of buying 5 shares indicates a strong buy action. In addition, since the reward function of the action-specialized expert model with extended action space is defined as multiplying reward value, m and extended action (the number of shares), the action-specialized expert model can obtain more various reward values, which have a wide range. Thus, if the model can detect the degree of obvious patterns, which can be the direction and magnitude of dynamic market movements from input state, then it can determine how many shares to buy or sell of a stock depending on the detected degree by choosing the correct extended action.

## Proposed ensemble model—action-specialized expert ensemble model

Fig 2 indicates the process of common ensemble model and our proposed model. The reward of common model is the raw value of profit or Sortino ratio and common ensemble consists of these models. The reward of our proposed model, on the other hand, is controlled by an additional value which is compensation for the expert action under specific condition. In this way, it consists of three different action-specialized expert models based on DRL. In Fig 2, the colored boxes represent enhanced expert action of each expert single model. In the common ensemble method, performance substantially improves because an ensemble of a plurality of networks can be averaged to reduce the deviation of the resulting network. Unlike the common ensemble method that combines similar models, our proposed ensemble method combines buy-, hold-, and sell-specialized single expert models to improve performance. For instance, our proposed ensemble model functions similarly to three experts from different fields cooperatively making decisions with unifying opinions. Thus, each expert model yields a different inference or decision with the same input; however, our ensemble method improves performance. When we employ it, we use the soft voting ensemble method, which can avoid

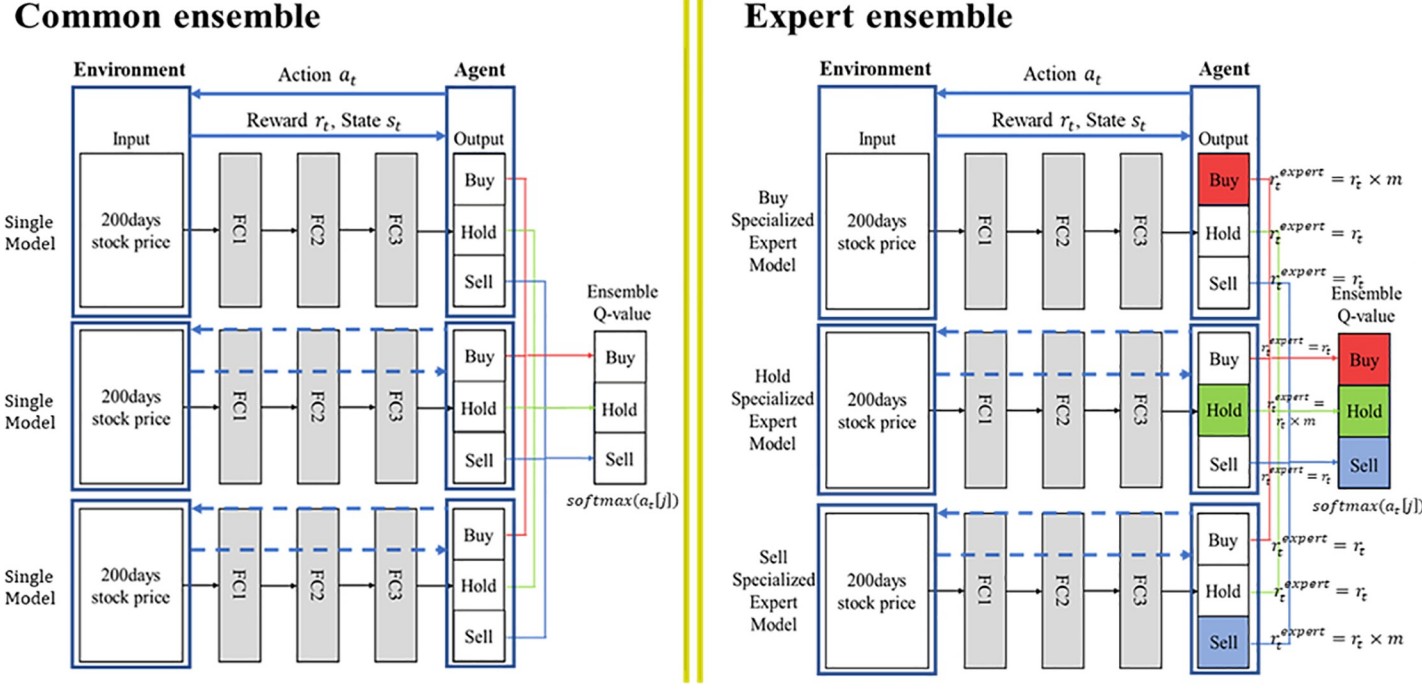

**Fig 2. Process of common ensemble model and our proposed model.**

loss of information [16]. The soft voting method equation is as follows.

$$Output_{tj} = softmax(a_t[j]) = \frac{a_t[j]}{\sum_{j=1}^{J} \exp(a_t[j])} \tag{11}$$

where $Output_{tj}$ is converted from the softmax function with $a_t[j]$ at time $t$. $a_t$ is the action as outputs of the model and $J$ is the number of outputs. The softmax function normalizes each action value, which is the Q-value in DQN in the expert model, between 0 and 1. Since the sum of all the action values after applying the softmax function becomes 1, each value of output layer of DQN in the expert model indicates the probability of each action. For the final decision of the expert ensemble model, the Q-value of DQN in each expert model takes the softmax function. After that, the average of the activated Q-values of buy, hold, and sell-specialized expert models become the final outputs of the expert ensemble model, that is, the Q-values of the expert ensemble model. Thus, the action of the highest Q-value of the expert ensemble model is selected as the final decision. To describe our proposed method in detail, the DQN algorithm for our model is provided in Algorithm 1 below.

**Algorithm 1** Deep Q Network for Single Models and Action Specialized Expert Models

**Hyperparameters:** *M*—size of experience replay memory, *R*—repeat step of training, $X_{train}$—training data set, $T_{train}$—episode of training data, *m*—predetermined positive constant, mini-batch size—64, *C*—episode of updating target $\hat{Q}$ network, $T_{test}$—episode of test data.

**Parameters:** *w*—weights of *Q* network, $w^-$—weights of target $\hat{Q}$ network

**Variables:** *Total Profit*—cumulative profit as performance measure, $s_t$—state space at time *t*, $a_t$—action space at time *t*, $r_t$—reward at time *t*, $r_t^{expert}$—reward for expert model at time *t*

Initialize replay memory *M*

Initialize the *Q* network with random weights *w*

```
Initialize the target Q̂ network with weights w⁻ = w
```
**Training Phase**
```
for STEP = 1, R do
  Set training data set Xtrain
  Total profit = 1
  for episode = 1, Ttrain do
    Set state st
    Choose action at following ε-greedy strategy in Q
    If common single model == True then
        rt = rt
    Else if action specialized expert model == True then
      If range range of profit == True and expert action == True then
        rt^expert = rt × m  (Eq (10))
    Else if range of profit == True and expert action == False then
        rt^expert = rt
    Else if range of profit == False and expert action == True then
        rt^expert = rt
    Else if range of profit == False and expert action == False then
        rt^expert = rt
    end if
  end if
  Set next state st+1
  If len(M) == max_memory then
    Remove the oldest memory from M
  Else
    Store memory (st, at, rt, st+1) in replay memory buffer M
  end if
  for each mini-batch sample from buffer M do
    Q(st, at) ← Q(st, at) + α · (rt+1 + γ maxa Q̂(st+1, a) − Q(st, at))
    Total profit ← Total profit+profitt.
  end for
  in every C episodes, reset Q̂ = Q, i.e., set weights w⁻ = w
  end for
end for
Clear replay memory buffer M
```
**Test Phase**
```
Set test data set Xtest for Online learning
Set Total profit =1
for episode = 1, Ttest do
  Repeat Training Phase with R = 1
end for
```
**Ensemble Phase**
```
Prepare three models of each expert action model
Ensemble these models by soft voting at each time t (Inference time
ensemble)
```

To prevent overfitting our proposed method and to train the network better, we used experience replay and epsilon-greedy in our DRL experiments. Regarding experience replay, all the experiences are saved in the replay memory in the shape of $<s_t, a_t, r_t, s_{t+1}>$ during the training of the DQN network. Then, the replay memory is uniformly shuffled to make a mini-batch of random samples so that the mini-batch sample is not sequential. This eliminates the time dependency of subsequent training samples. In addition, the observed experience is reused to train when it is sampled repeatedly and improves data usage efficiency. Thus, it helps to avoid local minima and prevent overfitting. Next, the epsilon-greedy method is used to solve exploration exploitation dilemmas in DRL. The epsilon-greedy method chooses an action randomly with probability $\varepsilon$ and the maximum Q-value action with probability (1-$\varepsilon$). The epsilon($\varepsilon$) is

decreased over an episode from 1 to 0.1. This will result in completely random moves to explore the state space maximally at the start of the training, which settles down to the fixed exploration rate of 0.1 at the end of the training. Therefore, the epsilon-greedy method helps to prevent overfitting or underfitting.

## Data

### Data design

In this study, we use the data of three indices: S&P500, HSI, and the Eurostoxx50, to verify our proposed method. We obtained these data from the Yahoo Finance Website and used the same period for each data set. Specifically, the training period spans from January 1987 to December 2006, and the test period from January 2007 to December 2017. By establishing the same time periods, we were able to compare how the RL model freely learns and yields different results over the same period in different environments. The data for the state space consists of the 200-days close price as the input, and the action space as the output generally relates to buy, hold, and sell, with 3, 11, and 21 actions according to the number of actions and experiments. The data set periods are described in Table 3. In order to discuss the trade-off between training costs and performance in more detail, we prepared three more training data sets with S&P500 with different time periods of 5, 10, and 15 years with same test data set period of 11 years. Based on these experimental settings, we could discuss the trade-off between different time period data set and the performance and another trade-off between training time and the performance.

### Data analysis

Fig 3 above demonstrates how each index changed during the same period. The graph indicates that, during the training period of 1987–2006, a common upward trend resulted from economic growth. Unlike the S&P500 and Eurostoxx50 movements, however, HSI displays different moves toward the end of 1990. During the test period, the three indices indicate different movements. In this period, S&P500 moved upward except during the global financial crisis in 2008; however, HSI and Eurostoxx50 exhibited large fluctuations even after the financial crisis. HSI recovered slightly after the drop; however, Eurostoxx50 failed to recover after the decrease. Against the backdrop of these differences, we can compare the effectiveness of RL in terms of training and showing results. The movement during the test period can be thought of as a Buy and Hold strategy [39]. In the test period, analyzing data with the Buy and Hold strategy indicates that S&P500 increased by 89% and HSI by 47.3%, while Eurostoxx50 decreased by 16.3% (-16.3%).

### Profit distribution

We analyzed our data set, and Fig 4 indicates profit distribution and data balance during the training period. Only training data were analyzed, and after this period, the model will update

**Table 3. Description of data sets' periods.**

| Index | Training | | Test | |
|---|---|---|---|---|
| | **Period** | **# of data** | **Period** | **# of data** |
| S&P500 | Jan 2, 1987–Dec 29, 2006 | 5040 | Jan 3, 2007–Dec 29, 2017 | 2767 |
| Hang Seng Index (HSI) | Jan 2, 1987–Dec 29, 2006 | 4935 | Jan 2, 2007–Dec 29, 2017 | 2710 |
| Eurostoxx50 | Jan 1, 1987–Dec 29, 2006 | 5151 | Jan 3, 2007–Dec 29, 2017 | 2745 |

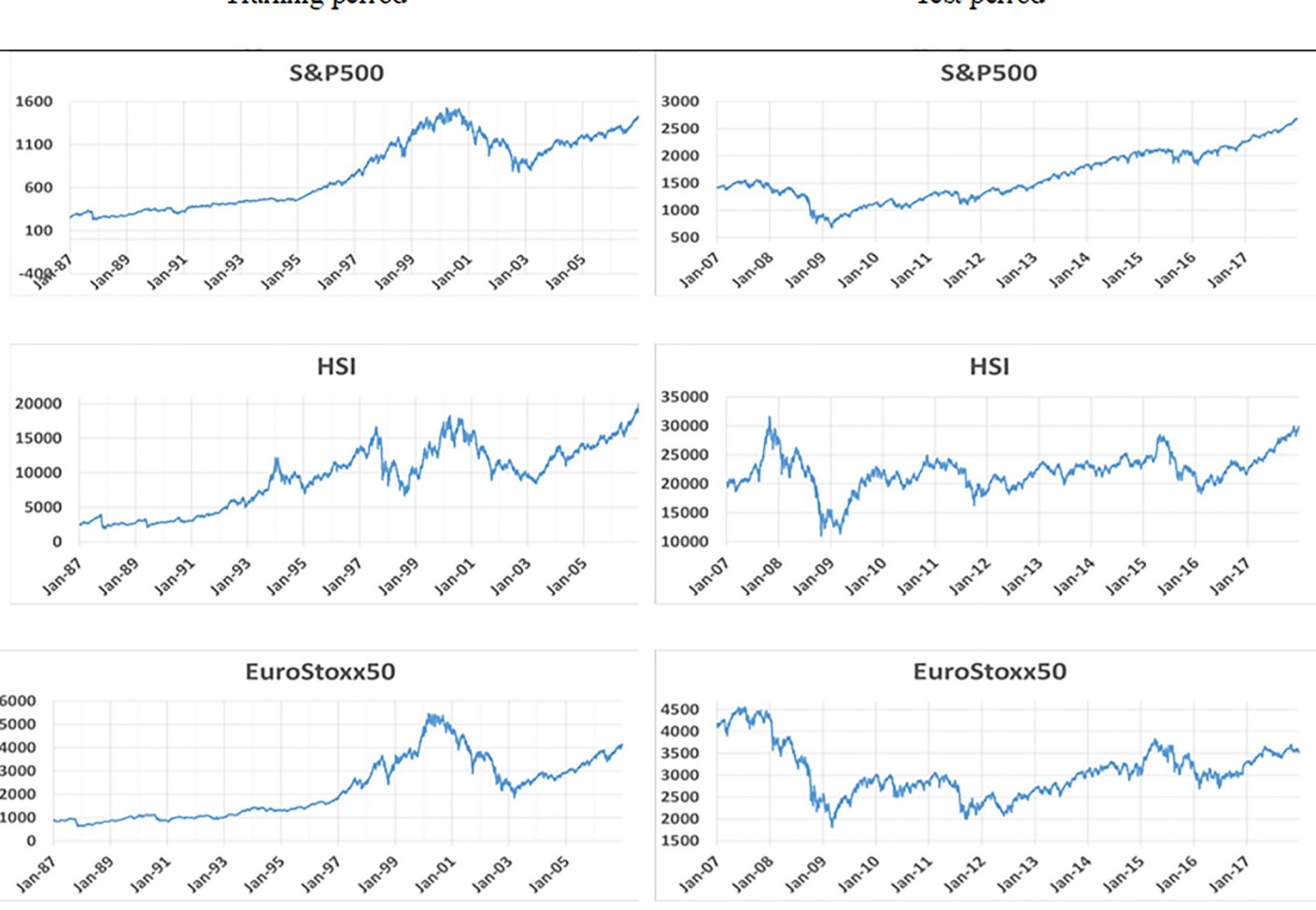

**Fig 3. Movements of the three indices in the training and test periods (Buy and Hold strategy).**

with test data using online learning. First, we divide the profit distribution into units of 0.5%, and all three indices seem to follow a normal distribution shape. Based on this analysis, the reward function was adjusted so that the action-specialized expert model could adaptively learn according to the profit. For example, the interval from 0.3% to 1%, which frequently occurs in the expert model for action of buy, is 3 times for the existing reward, 5 times for the interval from 1% to 2%, 6 times for the interval from 2% to 3%, 7 times for the interval from 3% to 5%, and 10 times for the interval that exceeds 5%. On the contrary, the adjusted part of the expert model for action of sell is the same, and the expert model for action of hold is applied 7 times in the -0.3% to 0.3% range. The reason for using 0.3% as the standard is that the transaction cost is assumed at 0.3% in many studies [40–44]. Therefore, our model learns for action of hold in the interval of less than 0.3%. The circle graphs for data balance also indicate the ratio of buy, hold, and sell data based on 0.3%, and data for the three indices appear to be balanced.

## Normality test

We attempted to interpret these three indices data sets by referring to their statistical properties in Table 4. As is known, stock returns are characterized by negative skewness and sharp

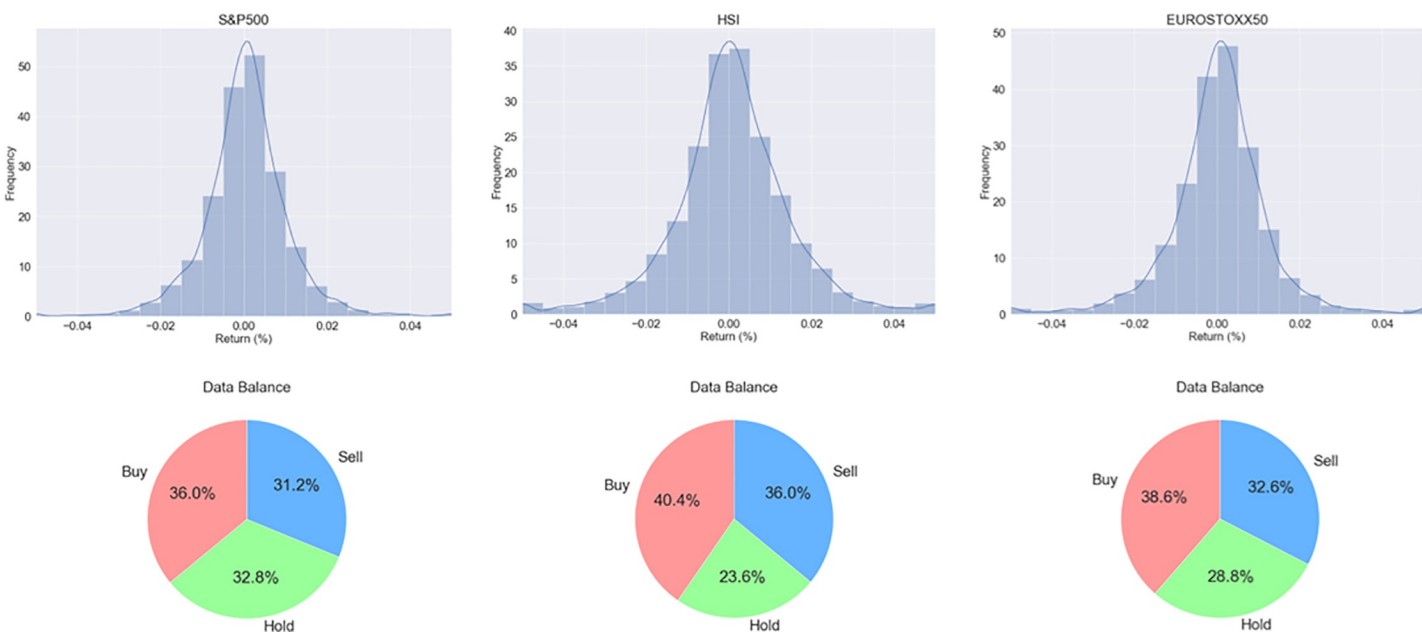

**Fig 4. Profit distribution during the training period and data balance for buy, hold, and sell.**

kurtosis. Negative skewness means a distribution shape where small profits are frequent but extreme losses occur. Sharp kurtosis is a characteristic of daily return distribution, and is lower during long-term return distribution.

Table 4 shows the basic descriptive statistics of each index data set. First, to analyze the distribution of train data set and test data set of three indices from Table 4, the train data set of S&P500 shows negative skewness of -1.48 and the test data set shows relatively weak negative skewness of -0.10. The train data set of HSI shows negative skewness of -1.94, while test data set shows positive skewness of 0.29. The train data set of Eurostoxx50 has a negative skewness of -0.17, which is relatively weaker than the other indices, and the train data set has a positive skewness of 0.12. Negative skewness is a distribution where small gains occur frequently and extreme losses occur, while positive skewness means distributions where small losses occur frequently but extreme gains occur. Considered together, all three indices show a more negative skewness of the train data set than the test data set, and an upward trend with statistical

**Table 4. Descriptive statistics of each index data sets.**

|  | S&P500 | | | HSI | | | STOXX50 | | |
|---|---|---|---|---|---|---|---|---|---|
|  | **Train** | **Test** | **Total** | **Train** | **Test** | **Total** | **Train** | **Test** | **Total** |
| **Count** | 5039 | 2767 | 7806 | 4934 | 2710 | 7644 | 5150 | 2745 | 7895 |
| **Mean** | 0.0004 | 0.0003 | 0.0004 | 0.0006 | 0.0003 | 0.0005 | 0.0004 | 0.0001 | 0.0003 |
| **Std** | 0.0107 | 0.0126 | 0.0114 | 0.0168 | 0.0159 | 0.0165 | 0.0122 | 0.0150 | 0.0132 |
| **Min** | -0.2047 | -0.0904 | -0.2047 | -0.3333 | -0.1270 | -0.3333 | -0.0793 | -0.0862 | -0.0862 |
| **25%** | -0.0046 | -0.0040 | -0.0044 | -0.0065 | -0.0068 | -0.0066 | -0.0051 | -0.0069 | -0.0058 |
| **50%** | 0.0005 | 0.0006 | 0.0006 | 0.0007 | 0.0005 | 0.0006 | 0.0007 | 0.0001 | 0.0005 |
| **75%** | 0.0056 | 0.0055 | 0.0056 | 0.0083 | 0.0079 | 0.0082 | 0.0063 | 0.0072 | 0.0066 |
| **Max** | 0.0910 | 0.1158 | 0.1158 | 0.1882 | 0.1435 | 0.1882 | 0.0733 | 0.1100 | 0.1100 |
| **Skewness** | -1.4796 | -0.1033 | -0.8337 | -1.9444 | 0.2875 | -1.2299 | -0.1703 | 0.1163 | -0.0354 |
| **Kurtosis** | 31.7070 | 11.1981 | 21.7192 | 45.6856 | 9.4156 | 34.5147 | 5.3473 | 5.8628 | 5.9805 |

properties from 1987 to 2007. In addition, the S&P500 with negative skewness in the test data set from 2007 to 2017 shows an upward trend graph while HSI with positive skewness shows an upward trend, but it is more volatile and lower rising than the S&P500 (S&P500: 87% increase, HSI: 47.3% increase). In addition, Eurostoxx50 shows a downward graph of -16.3%, which shows characteristics of positive skewness that are prone to frequent losses. Statistical characteristics of Kurtosis indicate the sharpness and the tail of the distribution. The kurtosis of the train data set of the S&P500 and HSI is 31.71 and 45.69, respectively, with a sharp normal distribution with a long tail. These test data sets are 11.20 and 9.42, respectively, and are more evenly distributed than the train data set. The kurtosis of Eurostoxx50 is 5.35 in the train data set and 5.86 in the test data set, which is relatively more evenly distributed than S&P500 and HSI. We can also check the volatility of each index in the Table 4, with the highest volatility in the order of HSI, Eurostoxx50, and S&P500.

Fig 5 indicates the quantile-quantile (Q-Q) plots for each index, and they do not demonstrate a linear pattern. This plot is a graphical technique for determining if two data sets come from populations with a common distribution. This indicates that the data set is not a normal distribution, and we establish that it follows a leptokurtic distribution shape through Figs 4 and 5.

## Experiments

### Experimental setup

The purpose of our proposed method is to create an automated trading system that is more profitable. To achieve this, we develop the action-specialized expert ensemble method with DRL. There are many studies to improve the accuracy of prediction in the financial sector, but a higher accuracy of prediction does not mean higher profit. Thus, without return or price prediction, we develop a profitable trading system based on DRL with action specific controlled reward function to create the ensemble model of action-specialized expert models different from the existing common ensemble method. The DQN network requires defining the state and action space of the problem, as well as a reward function.

### State space & action space

State space, as the input, uses the 200 days price data of each index. The agent analyzes the pattern over 200 days and learns to take action. These experimental environments are similar to those of previous studies [2, 7]. In this study, single and action-specialized expert models have the same state space.

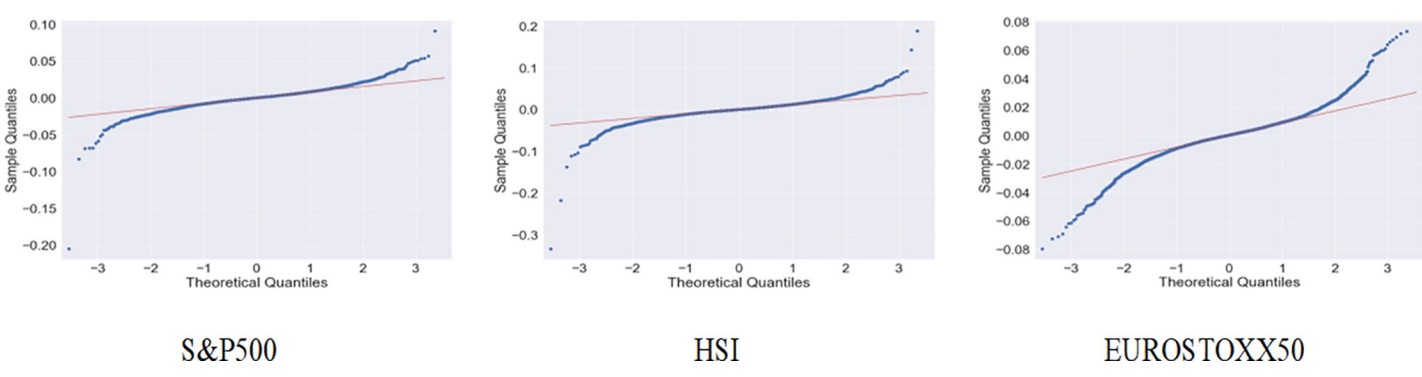

**Fig 5. Q-Q plots for each index.**

We attempt to apply the action space in three ways, and examine how results differ from those of existing experiments when the available quantity increased 5 and 10 times. Further, we want to verify our proposed method under various experimental situations. Therefore, the first experiment was conducted with the same 3-action method—buy, hold, and sell—in which the quantity of shares was limited to one. In other ways, we attempted to discretely increase buy and sell actions. If the action space has 11 actions, there are 5 actions for purchase, 5 actions for sale, and 1 hold action. Additionally, the 21-action space case has 10 actions for purchase, 10 actions for sale, and 1 hold action. Furthermore, these actions for purchase and sale are the number of shares ranging from 1 to 5 or 10. As in the state space, the action spaces of the single and expert models are the same.

### Reward function

In the single model, we applied three types of reward functions: the profit, Sharpe ratio, and Sortino ratio. We compared the trading system where only profit is used with the trading system where profit and volatility are used. We employed three types of action spaces: the 3-action, 11-action, and 21-action spaces. Thus, $a_t$ in Eqs (7), (8), and (9) is defined as the number of shares: $\{-1,0,1\}$,.$\{-5\sim-1,0,1\sim5\}$, and $\{-10\sim-1,0,1\sim10\}$. To be exact, since we cannot trade these indices in the real market, $a_t$ indicates the number of multiple shares of a stock. In Eq (7), $n$ is set to 100, and it is a network structure for maximizing the profit of 100 days by observing 200 days. To verify the sensitivity of our proposed model, we first use the profit function which only considers profit. Second, we use the Sharpe Ratio which allows for both profit and volatility. Third, we use the Sortino ratio which only considers volatility in loss.

### Action-specialized expert ensemble model

Our proposed model consists of three action-specialized expert models for buy, hold, and sell. Fig 6 shows the process of our proposed method on training, test, and ensemble phases. In the training phase, each action-specialized expert model for buy, hold, and sell is trained. Each expert model follows steps for the training phase in Algorithm 1. As aforementioned in Table 2, the action-specialized expert models have a specific range of profit, which is different from the DQN models, to control reward value. After the training phase, we performed online learning to match the RL approach with reflecting the dynamic financial market. We input the first mini-batch of test data to obtain the first outputs of each action-specialized expert model. Thereafter, through sliding window and the same mini-batch size as the test data, we trained each action-specialized expert model, including former test data again; we input this next mini-batch of test data to obtain the next outputs. The online learning continues till the end of the test data. Further, we compiled the three expert models by soft voting, described in section 3.4, at each time $t$ at inference time. To further explain the distinction between common and expert ensemble models, the common ensemble model requires three models, which are trained identically using the unenhanced reward function. The expert ensemble model, on the other hand, uses an enhanced reward function to create expert models for buy, hold, and sell action-specialized and ensemble these models.

Despite many advances in RL fields, DRL models are still unstable in learning, and hence, it is difficult to reproduce state-of-the-art performance. Therefore, Henderson et al. [45] suggest that presenting the mean and standard error of the five results shows a better performance of the model than only the topmost result. Accordingly, most recent DRL studies [45–49] present the mean and standard error (mean ± standard error) of five models. We ensembled the single models from the top five to three models and reselected the top five models from the ensemble results. Expert models were selected by combining each two models of buy, hold, and sell, and we selected

**Fig 6. Model training and the ensemble process on training as well as the test phase.**

the top five models again from the expert ensemble results. Therefore, we also trained the models of each approach 10 times and selected the top five to present the mean and standard error.

Fig 7 outlines the experimental steps. First, to compare the profit reward function with the Sharpe and Sortino ratios, we examine it with single models and noticed that two ratios were better than the profit reward function in two-thirds of these experiments. Second, since the Sortino ratio is slightly better than the Sharpe ratio in the results of single model experiments, we exclude the Sharpe ratio as a reward function. We then compare the profit reward function with the Sortino ratio, which jointly considers profit and volatility. We also collect the top five models of each reward function of the single model. Third, we train the action-specialized expert networks with the profit and Sortino ratio reward functions and then collect the top two models of each action-specialized expert model as Step 4. Step 5 is the ensemble step for single models. We ensemble these three models out of five single models and extract the top five out of the ten ensemble models according to the results. The next step is the development of action-specialized expert models. We establish each expert model out of two models and choose the top five of eight expert ensemble models. Lastly, we repeat Step 1 to 6 with three types of extended discrete action spaces and three types of index data sets.

Describing the detailed architecture of our Q-network, we use three hidden fully connected layers: the number of neurons is 200, 100, and 50, respectively. We use the ReLU activation function and the following hyperparameters where the replay memory ($M$) is 10000, the step of training ($R$) is 20, the episode of target Q-network ($C$) is 256, learning rate is 0.0001, the gamma($\gamma$) is the 0.85, and the mini batch size is 64; we also use Adam optimization.

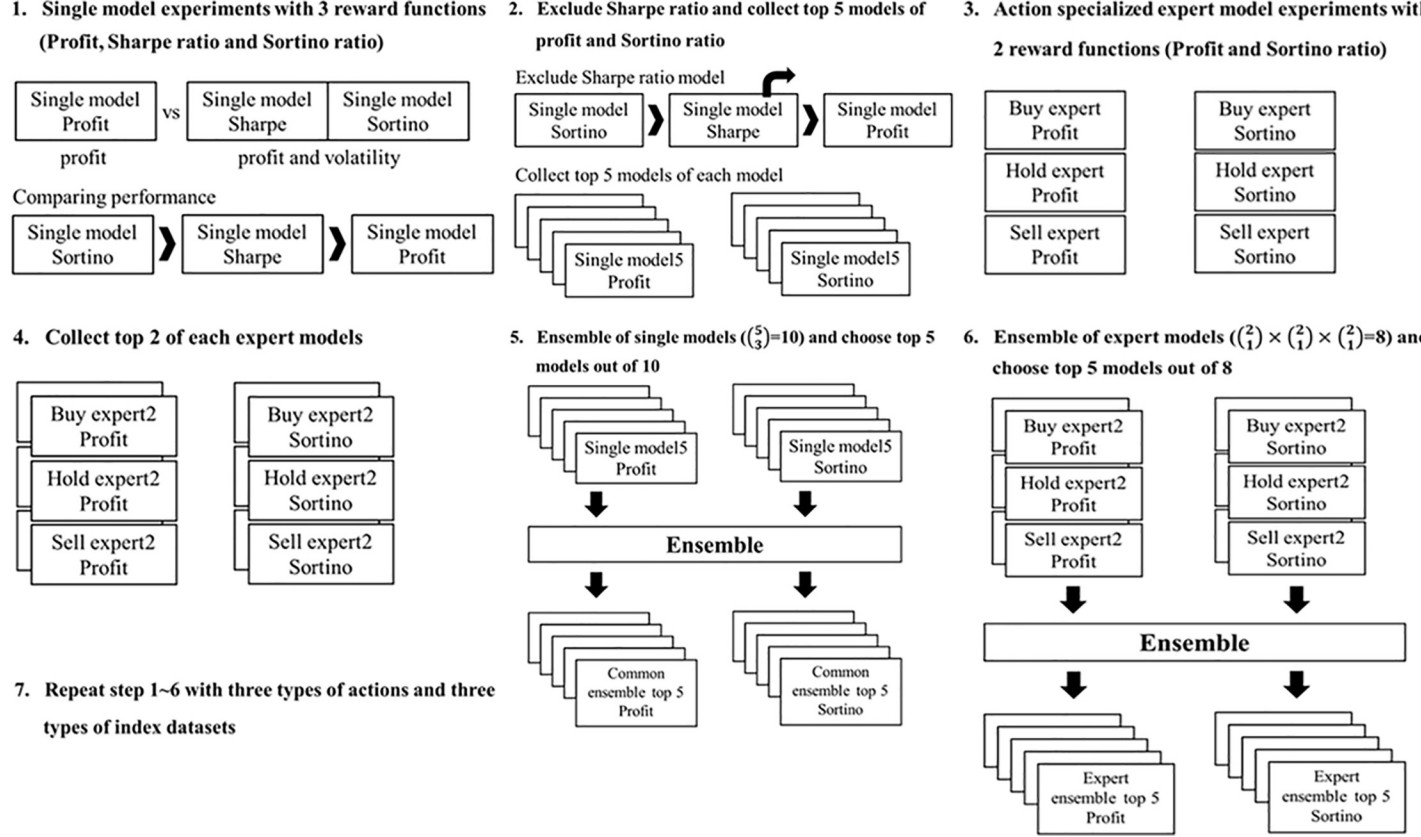

**Fig 7. The entire experiment processes.**

## Experimental results and discussion

### Single model with three reward functions

We investigated reward functions that consider both profit and volatility in the single model experiment. Table 5 summarizes the top five models of each single model result using the Sharpe and Sortino ratios as reward functions. In addition, the results of each ratio correlate with the profit reward function. Since the average or volatility of the return is different depending on the window size of time series data, we conducted various window size tests. However, we could not compare the two ratios even for each window size; consequently, we used a cross window average to compare them. Specifically, it was not possible to compare the experimental results to determine which of the two ratios is a better reward function depending on window size, action, or index. Since the Sortino ratio was slightly better than the Sharpe ratio as a result of cross-averaging, we only utilized the Sortino ratio when creating action-specialized expert models. We observe that the results of the two ratios are better than the result of profit in two-thirds of this experiment. We compared the reward functions using the profit and Sortino ratio in the following experimental outline.

### Results of action-specialized expert single model with two reward functions

Table 6 indicates results of the top two of each action-specialized expert models to be used in the final expert ensemble, and each single model as compared with an expert single model. When comparing results of the expert models according to each specific action, it is evident

**Table 5. Comparison of top five models' average profits of single models with different reward functions.**

**Top 5 average, S&P500**

| # of action | Ratio | Window size | | | | | | | Cross |
|---|---|---|---|---|---|---|---|---|---|
| | | **0** | **15** | **20** | **25** | **30** | **35** | **50** | **Average** |
| 3 | Profit | 3.516 ±0.131 | - | - | - | - | - | - | - |
| | Sharpe | - | 3.133 ±0.050 | 3.192 ±0.098 | 3.159 ±0.035 | 3.178 ±0.146 | 3.203 ±0.094 | 3.242 ±0.047 | 3.184 ±0034 |
| | Sortino | - | 3.133 ±0.103 | 3.212 ±0.127 | 3.156 ±0.144 | 3.153 ±0.063 | 3.258 ±0.110 | 3.247 ±0.068 | 3.193 ±0.048 |
| 11 | Profit | 8.958 ±0.584 | - | - | - | - | - | - | - |
| | Sharpe | - | 10.369 ±0.252 | 10.654 ±0.787 | 10.402 ±0.337 | 11.907 ±0.420 | 10.328 ±0.674 | 10.274 ±0.241 | 10.656 ±0.572 |
| | Sortino | - | 10.272 ±0.588 | 10.455 ±0.402 | 10.647 ±0.634 | 10.351 ±0.057 | 10.849 ±0.560 | 10.516 ±0.337 | 10.515 ±0.191 |
| 21 | Profit | 20.71 ±0.424 | - | - | - | - | - | - | - |
| | Sharpe | - | 20.13 ±0.341 | 19.992 ±0.362 | 20.006 ±0.974 | 20.001 ±0.621 | 19.476 ±0.252 | 20.583 ±0.385 | 20.031 ±0.323 |
| | Sortino | - | 20.056 ±0.472 | 20.119 ±0.386 | 20.098 ±0.937 | 20.223 ±0.614 | 19.898 ±0.231 | 22.584 ±0.758 | 20.496 ±0.939 |

**Top 5 average, HSI**

| # of action | Ratio | 0 | 15 | 20 | 25 | 30 | 35 | 50 | Average |
|---|---|---|---|---|---|---|---|---|---|
| 3 | Profit | 3.22 ±0.032 | - | - | - | - | - | - | - |
| | Sharpe | - | 3.576 ±0.032 | 3.486 ±0.096 | 3.426 ±0.057 | 3.443 ±0.116 | 3.537 ±0.064 | 3.488 ±0.092 | 3.493 ±0.051 |
| | Sortino | - | 3.281 ±0.129 | 3.574 ±0.116 | 3.503 ±0.062 | 3.525 ±0.157 | 3.667 ±0.069 | 3.554 ±0.088 | 3.517 ±0.118 |
| 11 | Profit | 10.952 ±0.167 | - | - | - | - | - | - | - |
| | Sharpe | - | 11.968 ±0.508 | 12.117 ±0.437 | 12.175 ±0.305 | 11.828 ±0.459 | 11.411 ±0.427 | 11.598 ±0.617 | 11.85 ±0.273 |
| | Sortino | - | 11.446 ±0.330 | 11.75 ±0.194 | 12.156 ±0.363 | 11.319 ±0.251 | 11.801 ±0.271 | 11.846 ±0.544 | 11.72 ±0.274 |
| 21 | Profit | 21.436 ±1.165 | - | - | - | - | - | - | - |
| | Sharpe | - | 23.514 ±0.270 | 22.393 ±0.460 | 22.269 ±0.224 | 22.476 ±1.008 | 22.924 ±0.511 | 23.129 ±0.631 | 22.784 ±0.445 |
| | Sortino | - | 23.313 ±0.654 | 22.31 ±0.362 | 23.267 ±0.300 | 23.29 ±0.363 | 22.532 ±0.867 | 23.456 ±0.322 | 23.028 ±0.438 |

**Top 5 average, Eurostoxx50**

| # of action | Ratio | 0 | 15 | 20 | 25 | 30 | 35 | 50 | Average |
|---|---|---|---|---|---|---|---|---|---|
| 3 | Profit | 3.379 ±0.093 | - | - | - | - | - | - | - |
| | Sharpe | - | 3.456 ±0.096 | 3.415 ±0.118 | 3.425 ±0.174 | 3.553 ±0.202 | 3.319 ±0.028 | 3.284 ±0.025 | 3.409 ±0.088 |
| | Sortino | - | 3.556 ±0.205 | 3.265 ±0.112 | 3.389 ±0.017 | 3.314 ±0.053 | 3.327 ±0.105 | 3.415 ±0.125 | 3.378 ±0.094 |
| 11 | Profit | 9.068 ±0.127 | - | - | - | - | - | - | - |
| | Sharpe | - | 10.883 ±0.408 | 11.942 ±0.685 | 10.947 ±0.179 | 10.906 ±0.088 | 11.233 ±0.851 | 11.645 ±0.687 | 11.259 ±0.404 |
| | Sortino | - | 11.323 ±0.370 | 10.591 ±0.110 | 10.704 ±0.364 | 12.412 ±0.167 | 11.219 ±0.554 | 11.497 ±0.559 | 11.291 ±0.597 |
| 21 | Profit | 22.99 ±0.740 | - | - | - | - | - | - | - |
| | Sharpe | - | 20.649 ±0.405 | 23.793 ±1.162 | 21.978 ±0.301 | 22.241 ±1.184 | 20.651 ±0.272 | 21.049 ±0.536 | 21.727 ±1.109 |
| | Sortino | - | 21.258 ±1.055 | 22.624 ±0.694 | 21.778 ±0.424 | 23.255 ±0.843 | 22.325 ±0.850 | 21.098 ±0.455 | 22.056 ±0.760 |

that profit results of a few actions are lower than others in each expert model. For example, the profit yielded by expert models for buy and hold is similar, while the profit yielded by the expert model for sell is relatively low. In addition, the expert model for buy is better than results of expert models for hold and sell. From these results, we noticed that the Sortino ratio results were better than profit results in two-thirds of the single experiment. However, it is unclear whether expert models may be effectively compared to single models. In other words, the experiment demonstrates different results by index, number of actions, specific action, and reward function. To demonstrate this comparison, we include the results in the Table 6.

## Results of the action-specialized expert ensemble model

We investigated results of the proposed action-specialized expert ensemble system. We compared ensemble results of expert models, which are specialized for each action, to ensemble results of single models, which are well-balanced learning methods. Tables 7–9 and Figs 8–10 indicate results of the top five models of all experiments. Comparisons of ensemble methods

**Table 6. Comparison of top 2 action-specialized expert models and single models.**

**Top 2 returns of expert and single model, S&P500**

| # of action | # of model | Single Profit | Profit Expert | | | Single Sortino | Sortino Expert | | |
|---|---|---|---|---|---|---|---|---|---|
| | | | Buy | Hold | Sell | | Buy | Hold | Sell |
| 3 | 1 | 3.774 | 3.657 | 3.709 | 3.089 | 3.242 | 3.563 | 3.521 | 3.305 |
| | 2 | 3.492 | 3.473 | 3.252 | 2.913 | 3.217 | 3.481 | 3.396 | 3.269 |
| 11 | 1 | 9.899 | 11.376 | 11.056 | 9.297 | 10.43 | 11.395 | 9.907 | 9.612 |
| | 2 | 9.066 | 11.049 | 10.624 | 9.189 | 10.393 | 11.154 | 9.848 | 9.569 |
| 21 | 1 | 21.482 | 22.266 | 19.377 | 22.296 | 21.237 | 19.314 | 19.491 | 22.047 |
| | 2 | 20.85 | 21.488 | 18.59 | 21.502 | 20.43 | 18.939 | 19.056 | 20.322 |

**Top 2 returns of expert and single model, HSI**

| # of action | # of model | Single Profit | Buy | Hold | Sell | Single Sortino | Buy | Hold | Sell |
|---|---|---|---|---|---|---|---|---|---|
| 3 | 1 | 3.269 | 3.625 | 3.433 | 3.71 | 3.555 | 3.767 | 3.552 | 3.498 |
| | 2 | 3.232 | 3.436 | 3.281 | 3.483 | 3.504 | 3.426 | 3.378 | 3.204 |
| 11 | 1 | 11.196 | 10.421 | 10.775 | 10.771 | 11.686 | 10.17 | 10.901 | 11.972 |
| | 2 | 11.108 | 10.102 | 10.401 | 10.624 | 11.494 | 10.1 | 10.804 | 11.521 |
| 21 | 1 | 22.947 | 19.18 | 21.034 | 20.011 | 23.892 | 24.41 | 23.438 | 22.718 |
| | 2 | 22.764 | 18.943 | 20.921 | 19.813 | 23.427 | 23.84 | 23.099 | 22.066 |

**Top 2 returns of expert and single model, Eurostoxx50**

| # of action | # of model | Single Profit | Buy | Hold | Sell | Single Sortino | Buy | Hold | Sell |
|---|---|---|---|---|---|---|---|---|---|
| 3 | 1 | 3.479 | 3.467 | 3.549 | 3.209 | 3.413 | 3.573 | 3.296 | 3.302 |
| | 2 | 3.473 | 3.363 | 3.411 | 3.19 | 3.323 | 3.384 | 3.255 | 3.174 |
| 11 | 1 | 9.291 | 9.837 | 9.921 | 9.945 | 12.628 | 9.883 | 10.857 | 10.807 |
| | 2 | 9.09 | 9.575 | 9.796 | 9.783 | 12.593 | 9.462 | 10.561 | 10.801 |
| 21 | 1 | 24.395 | 24.544 | 21.074 | 20.831 | 22.729 | 23.951 | 21.558 | 22.225 |
| | 2 | 23.038 | 23.48 | 20.919 | 20.332 | 22.229 | 22.287 | 20.646 | 21.477 |

**Table 7. Cumulative profits of top five models on S&P500.**

| # of action | # of model | Reward function and Ensemble | | | | | |
|---|---|---|---|---|---|---|---|
| | | Profit | PE | EPE | Sortino | SE | ESE |
| 3 | 1 | 3.774 | 4.093 | 4.394 | 3.242 | 4.142 | 4.499 |
| | 2 | 3.492 | 4.016 | 4.36 | 3.217 | 3.781 | 4.488 |
| | 3 | 3.446 | 4.003 | 4.221 | 3.104 | 3.757 | 4.471 |
| | 4 | 3.438 | 3.85 | 4.153 | 3.102 | 3.669 | 4.443 |
| | 5 | 3.43 | 3.792 | 4.106 | 3.1 | 3.655 | 4.423 |
| | Avg | 3.516 ±0.131 | 3.951 ±0.112 | 4.247 ±0.113 | 3.153 ±0.063 | 3.801 ±0.177 | 4.465 ±0.028 |
| 11 | 1 | 9.899 | 10.145 | 17.059 | 10.43 | 13.02 | 16.591 |
| | 2 | 9.066 | 9.942 | 16.602 | 10.393 | 12.589 | 15.829 |
| | 3 | 9.06 | 9.912 | 16.178 | 10.346 | 12.197 | 15.821 |
| | 4 | 8.64 | 9.878 | 16.083 | 10.32 | 12.038 | 15.795 |
| | 5 | 8.123 | 9.736 | 15.97 | 10.267 | 11.997 | 15.68 |
| | Avg | 8.958 ±0.584 | 9.923 ±0.132 | 16.378 ±0.402 | 10.351 ±0.057 | 12.368 ±0.387 | 15.943 ±0.328 |
| 21 | 1 | 21.482 | 24.564 | 28.346 | 21.237 | 23.457 | 26.926 |
| | 2 | 20.85 | 24.431 | 26.926 | 20.43 | 23.267 | 26.573 |
| | 3 | 20.46 | 24.331 | 26.837 | 20.267 | 21.88 | 26.292 |
| | 4 | 20.401 | 23.835 | 26.576 | 19.659 | 21.843 | 26.258 |
| | 5 | 20.359 | 23.626 | 26.499 | 19.522 | 21.458 | 26.14 |
| | Avg | 20.71 ±0.424 | 24.158 ±0.363 | 27.037 ±0.673 | 20.223 ±0.613 | 22.381 ±0.817 | 26.438 ±0.282 |

PE: profit ensemble (common ensemble), EPE: expert profit ensemble, SE: Sortino ensemble (common ensemble), ESE: expert Sortino ensemble

**Table 8. Cumulative profits of top five models on HSI.**

| # of action | # of model | Reward function and Ensemble | | | | | |
|---|---|---|---|---|---|---|---|
| | | **Profit** | **PE** | **EPE** | **Sortino** | **SE** | **ESE** |
| **3** | 1 | 3.269 | 3.833 | 4.68 | 3.555 | 4.02 | 4.516 |
| | 2 | 3.232 | 3.74 | 4.662 | 3.504 | 3.888 | 4.319 |
| | 3 | 3.216 | 3.699 | 4.644 | 3.383 | 3.855 | 4.302 |
| | 4 | 3.215 | 3.684 | 4.501 | 3.376 | 3.813 | 4.246 |
| | 5 | 3.17 | 3.643 | 4.465 | 3.345 | 3.8 | 4.231 |
| | Avg | 3.22 ±0.032 | 3.72 ±0.065 | 4.59 ±0.089 | 3.432 ±0.082 | 3.875 ±0.079 | 4.323 ±0.102 |
| **11** | 1 | 11.196 | 14.05 | 18.102 | 11.686 | 14.462 | 17.875 |
| | 2 | 11.108 | 13.985 | 16.62 | 11.494 | 14.45 | 17.106 |
| | 3 | 10.847 | 13.92 | 16.22 | 11.32 | 14.158 | 16.997 |
| | 4 | 10.833 | 13.783 | 16.028 | 11.08 | 14.117 | 16.963 |
| | 5 | 10.778 | 13.771 | 15.862 | 11.014 | 13.872 | 16.921 |
| | Avg | 10.952 ±0.167 | 13.902 ±0.110 | 16.566 ±0.808 | 11.319 ±0.251 | 14.212 ±0.222 | 17.172 ±0.357 |
| **21** | 1 | 22.947 | 24.277 | 26.835 | 23.892 | 26.355 | 31.558 |
| | 2 | 22.764 | 24.265 | 26.342 | 23.427 | 25.929 | 31 |
| | 3 | 20.655 | 23.934 | 25.141 | 23.226 | 25.582 | 30.158 |
| | 4 | 20.431 | 22.249 | 23.752 | 23.099 | 25.465 | 29.759 |
| | 5 | 20.382 | 22.215 | 22.873 | 22.805 | 25.046 | 29.521 |
| | Avg | 21.436 ±1.164 | 23.388 ±0.952 | 24.989 ±1.501 | 23.29 ±0.362 | 25.675 ±0.442 | 30.399 ±0.767 |

are conducted using two reward functions (profit and Sortino ratio). First, the subsequent tables are analyzed from the perspective of the expert ensemble, common ensemble, and single models.

**Table 9. Cumulative profits of top five models on Eurostoxx50.**

| # of action | # of model | Reward function and Ensemble | | | | | |
|---|---|---|---|---|---|---|---|
| | | **Profit** | **PE** | **EPE** | **Sortino** | **SE** | **ESE** |
| **3** | 1 | 3.479 | 3.768 | 4.348 | 3.413 | 3.776 | 4.124 |
| | 2 | 3.473 | 3.747 | 4.278 | 3.323 | 3.657 | 4.112 |
| | 3 | 3.398 | 3.717 | 4.265 | 3.283 | 3.577 | 4.103 |
| | 4 | 3.298 | 3.635 | 4.256 | 3.281 | 3.571 | 4.099 |
| | 5 | 3.248 | 3.582 | 4.194 | 3.271 | 3.566 | 4.006 |
| | Avg | 3.379 ±0.093 | 3.69 ±0.070 | 4.268 ±0.049 | 3.314 ±0.052 | 3.629 ±0.081 | 4.089 ±0.042 |
| **11** | 1 | 9.291 | 10.025 | 16.419 | 12.628 | 15.001 | 16.686 |
| | 2 | 9.09 | 9.866 | 15.746 | 12.593 | 14.262 | 16.669 |
| | 3 | 9.064 | 9.681 | 15.628 | 12.319 | 14.203 | 15.736 |
| | 4 | 8.97 | 9.613 | 15.566 | 12.306 | 14.126 | 15.673 |
| | 5 | 8.925 | 9.581 | 15.207 | 12.213 | 13.871 | 15.399 |
| | Avg | 9.068 ±0.127 | 9.753 ±0.168 | 15.713 ±0.396 | 12.412 ±0.167 | 14.293 ±0.378 | 16.033 ±0.539 |
| **21** | 1 | 24.395 | 27.238 | 34.863 | 22.729 | 27.526 | 30.05 |
| | 2 | 23.038 | 24.646 | 32.15 | 22.229 | 26.785 | 29.418 |
| | 3 | 22.629 | 24.57 | 31.965 | 22.209 | 26.781 | 29.129 |
| | 4 | 22.575 | 24.391 | 31.959 | 21.747 | 26.766 | 29.051 |
| | 5 | 22.311 | 24.304 | 31.69 | 21.669 | 26.743 | 27.763 |
| | Avg | 22.99 ±0.740 | 25.03 ±1.110 | 32.525 ±1.178 | 22.117 ±0.383 | 26.92 ±0.303 | 29.082 ±0.747 |

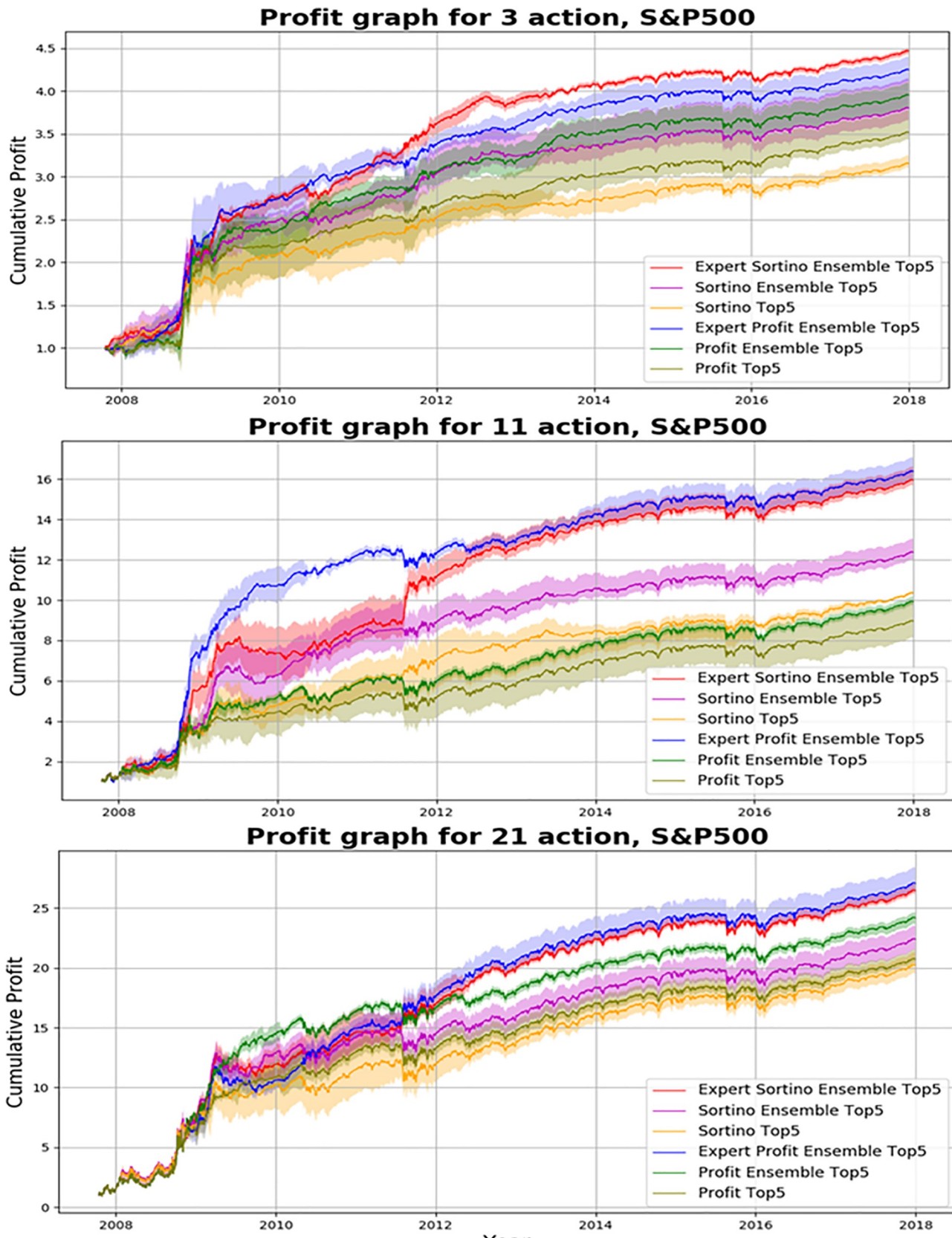

**Fig 8. Performance of DQN, common ensemble, and our proposed model with two reward functions on S&P500.**

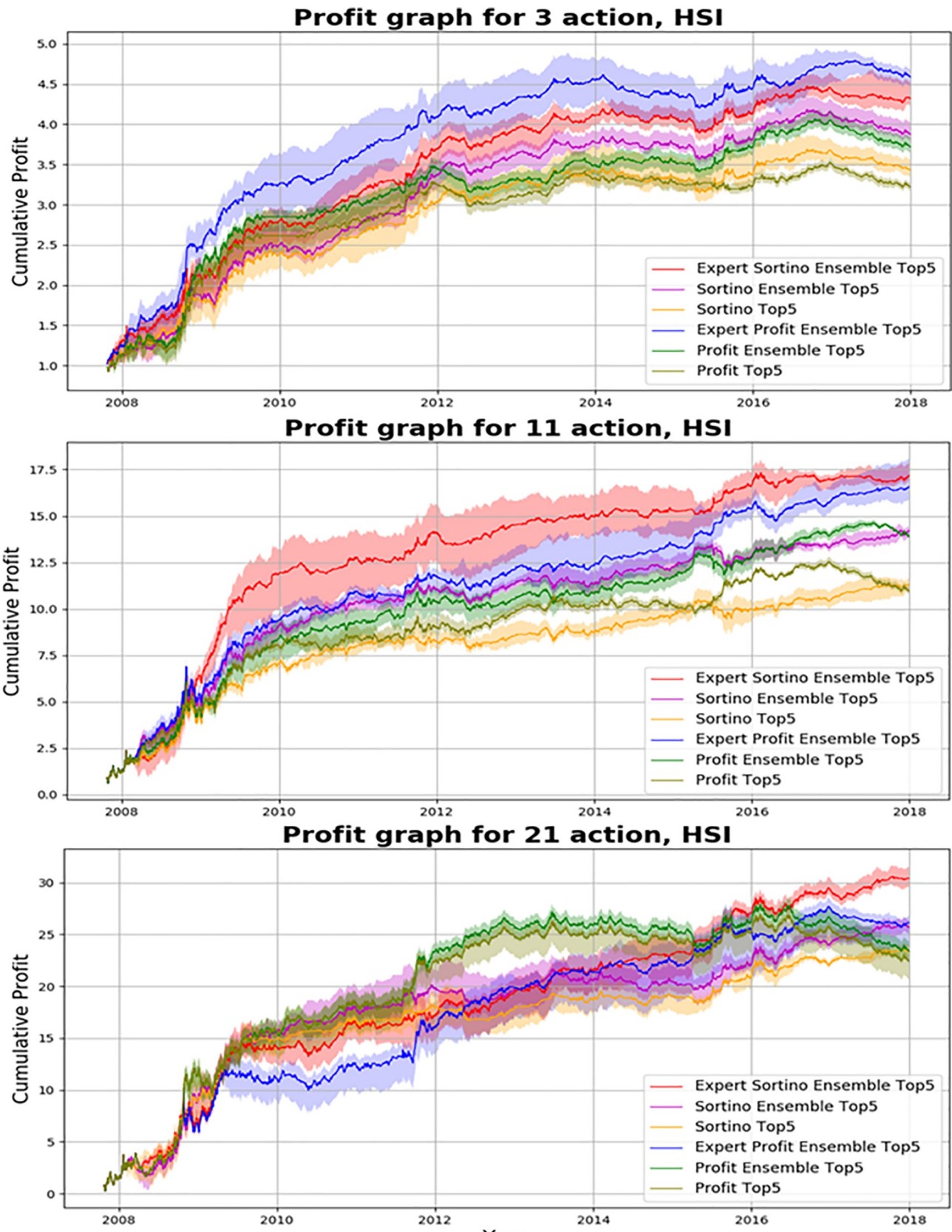

**Fig 9. Performance of DQN, common ensemble, and our proposed model with two reward functions on Hang Seng Index.**

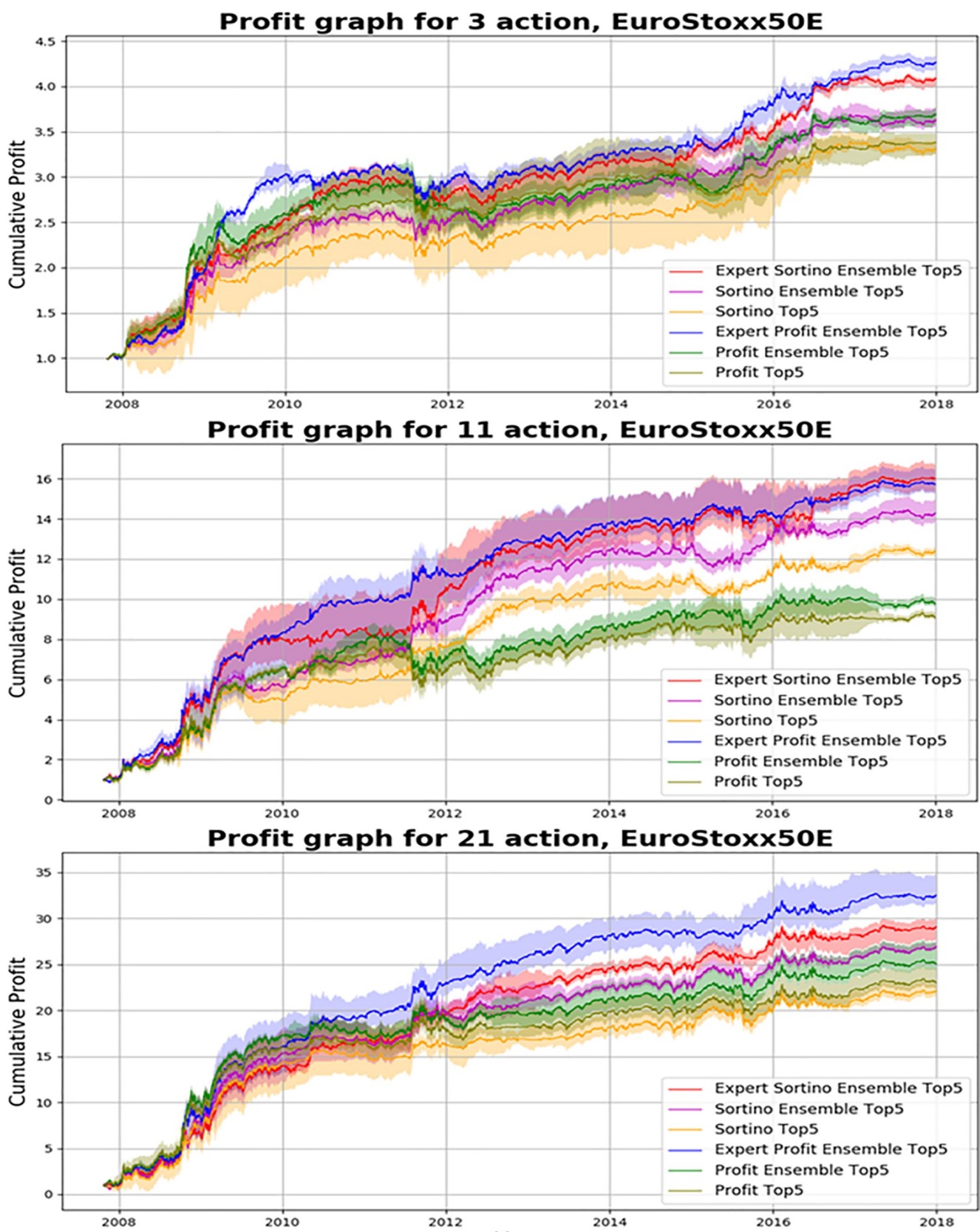

**Fig 10. Performance of DQN, common ensemble, and our proposed model with two reward functions on Eurostoxx50.**

Table 10 summarizes the top five averages for comparing expert ensemble methods to common ensemble and single models. As a result of analyzing the application of each ensemble, the increase in the range of the models (PE, SE) applying the common ensemble in the single model (Profit, Sortino) was 7.6–26.9%, which indicated an average increase of 14.6%. On the other hand, the profit range of the model (EPE, ESE) applying an action-specialized expert ensemble was 16.6–82.8%, which was 39.1% on an average. As a result, our proposed expert ensemble method in this study was 21.6% more effective than the common ensemble method. The ensemble method is widely used in general machine and deep learning; however, there are only a few cases applied in RL. Likewise, since the experimental results of the ensemble method as a newly attempted expert model appeared to be effective, we expect that our proposed method can be applied in further expansion of financial and various fields in the future.

Figs 8–10 indicate the average of the top five models, demonstrated by the thick colored line, and their standard error. Our proposed action-specialized expert ensemble model is most effective for the profit and Sortino ratio reward functions. In certain cases, the blue line—representing the expert ensemble with profit—is higher than the red line, which represents the ensemble model with the Sortino ratio. Regardless, our experiments indicate consistent results with the single, common ensemble, and expert ensemble models.

## Experimental results of the extended discrete action space

We examine robustness of the proposed ensemble system with the extension of the discrete action space. Above all, if we analyze Table 11, the increase in rate of the 3-action to the 11-action space is 302.4–505.5%, and the average increase is 427.2%. When increased from the 3-action to the 21-action space, the rate of the increase ranged from 668.1% to 985.8%, with an

**Table 10. Average profit and increasing rate with common and expert ensemble.**

| Average profit and increasing rate, S&P500 | | | | | | | |
|---|---|---|---|---|---|---|---|
| # of action | | Reward function and Ensemble | | | | | |
| | | **Profit** | **PE** | **EPE** | **Sortino** | **SE** | **ESE** |
| 3 | Avg | 3.516 ±0.131 | 3.951 ±0.112 | 4.247 ±0.113 | 3.153 ±0.063 | 3.801 ±0.177 | 4.465 ±0.028 |
| | % | - | 12.4 | 20.8 | - | 20.5 | 41.6 |
| 11 | Avg | 8.958 ±0.584 | 9.923 ±0.132 | 16.378 ±0.402 | 10.351 ±0.057 | 12.368 ±0.387 | 15.943 ±0.328 |
| | % | - | 10.8 | 82.8 | - | 19.5 | 54.0 |
| 21 | Avg | 20.71 ±0.424 | 24.158 ±0.363 | 27.037 ±0.673 | 20.223 ±0.613 | 22.381 ±0.817 | 26.438 ±0.282 |
| | % | - | 16.6 | 30.5 | - | 10.7 | 30.7 |
| Average profit and increasing rate, HSI | | | | | | | |
| 3 | Avg | 3.22 ±0.032 | 3.72 ±0.065 | 4.59 ±0.089 | 3.432 ±0.082 | 3.875 ±0.079 | 4.323 ±0.102 |
| | % | - | 15.5 | 42.6 | - | 12.9 | 25.9 |
| 11 | Avg | 10.952 ±0.167 | 13.902 ±0.110 | 16.566 ±0.808 | 11.319 ±0.251 | 14.212 ±0.222 | 17.172 ±0.357 |
| | % | - | 26.9 | 51.3 | - | 25.6 | 51.7 |
| 21 | Avg | 21.436 ±1.164 | 23.388 ±0.952 | 24.989 ±1.501 | 23.29 ±0.362 | 25.675 ±0.442 | 30.399 ±0.767 |
| | % | - | 9.1 | 16.6 | - | 10.2 | 30.5 |
| Average profit and increasing rate, Eurostoxx50 | | | | | | | |
| 3 | Avg | 3.379 ±0.093 | 3.69 ±0.070 | 4.268 ±0.049 | 3.314 ±0.052 | 3.629 ±0.081 | 4.089 ±0.042 |
| | % | - | 9.2 | 26.3 | - | 9.5 | 23.4 |
| 11 | Avg | 9.068 ±0.127 | 9.753 ±0.168 | 15.713 ±0.396 | 12.412 ±0.167 | 14.293 ±0.378 | 16.033 ±0.539 |
| | % | - | 7.6 | 73.3 | - | 15.2 | 29.2 |
| 21 | Avg | 22.99 ±0.740 | 25.03 ±1.110 | 32.525 ±1.178 | 22.117 ±0.383 | 26.92 ±0.303 | 29.082 ±0.747 |
| | % | - | 8.9 | 41.5 | - | 21.7 | 31.5 |

**Table 11. Increasing rate of profit by extended action on each index.**

| Increasing rate of profit by extended action on S&P500 | | | | | | | |
|---|---|---|---|---|---|---|---|
| # of action | | Reward function and Ensemble | | | | | |
| | | Profit | PE | EPE | Sortino | SE | ESE |
| 3 | Avg | 3.516 ±0.131 | 3.951 ±0.112 | 4.247 ±0.113 | 3.153 ±0.063 | 3.801 ±0.177 | 4.465 ±0.028 |
| 11 | Avg | 8.958 ±0.584 | 9.923 ±0.132 | 16.378 ±0.402 | 10.351 ±0.057 | 12.368 ±0.387 | 15.943 ±0.328 |
| | % | 316.3 | 302.4 | 473.6 | 434.3 | 405.9 | 431.3 |
| 21 | Avg | 20.71 ±0.424 | 24.158 ±0.363 | 27.037 ±0.673 | 20.223 ±0.613 | 22.381 ±0.817 | 26.438 ±0.282 |
| | % | 783.4 | 784.8 | 801.9 | 892.8 | 763.4 | 734.2 |
| Increasing rate of profit by extended action on HSI | | | | | | | |
| 3 | Avg | 3.22 ±0.032 | 3.72 ±0.065 | 4.59 ±0.089 | 3.432 ±0.082 | 3.875 ±0.079 | 4.323 ±0.102 |
| 11 | Avg | 10.952 ±0.167 | 13.902 ±0.110 | 16.566 ±0.808 | 11.319 ±0.251 | 14.212 ±0.222 | 17.172 ±0.357 |
| | % | 448.3 | 474.4 | 433.6 | 424.2 | 459.5 | 486.7 |
| 21 | Avg | 21.436 ±1.164 | 23.388 ±0.952 | 24.989 ±1.501 | 23.29 ±0.362 | 25.675 ±0.442 | 30.399 ±0.767 |
| | % | 920.5 | 823.2 | 668.1 | 916.3 | 858.2 | 884.8 |
| Increasing rate of profit by extended action on Eurostoxx50 | | | | | | | |
| 3 | Avg | 3.379 ±0.093 | 3.69 ±0.070 | 4.268 ±0.049 | 3.314 ±0.052 | 3.629 ±0.081 | 4.089 ±0.042 |
| 11 | Avg | 9.068 ±0.127 | 9.753 ±0.168 | 15.713 ±0.396 | 12.412 ±0.167 | 14.293 ±0.378 | 16.033 ±0.539 |
| | % | 339.1 | 325.4 | 450.2 | 493.1 | 505.5 | 486.7 |
| 21 | Avg | 22.99 ±0.740 | 25.03 ±1.110 | 32.525 ±1.178 | 22.117 ±0.383 | 26.92 ±0.303 | 29.082 ±0.747 |
| | % | 924.2 | 893.4 | 964.7 | 912.5 | 985.8 | 909.2 |

average of 856.7%. We calculate these numbers only for increasing rate from the base amount of investment 1.0 because of the different number of shares. For example, "302.4%" is calculated by the profit rate of PE of 11-action divide by profit rate of PE of 3-action (e.g., $(9.923 - 1.0)/(3.351 - 1.0) = 3.024$). In simple comparison, the number of multiple shares of a stock increased 5-fold when increasing from the existing 3-action to the 11-action space. In addition, upon the increase from the 3-action to the 21-action space, the quantity increases by 10 times. However, as the number of actions increases, it is possible to select under 5 or 10 actions and experimentally obtain a mean value smaller than the maximum expected value. Unlike selecting the quantity for minimum 0 or maximum 1 in the 3-action space in network learning, the model will choose the quantity in a flexible way, like 1 to 5 or 1 to 10 in the 11-action and 21-action space, respectively.

Further, we perform an extra test to compare the extended discrete actions in the 11-action and 21-action spaces with multiple shares of a stock in the 3-action space. Fig 11 displays average performance of extended discrete actions and multi shares on each index. First, to explain the *x*-axis, the 3-action space is for buy, hold, and sell with 1 share. Next, the 3-action space with 5 shares is for buy, hold, and sell with 5 shares. The following 11-action space is for 5 actions for buy from 1 to 5 shares, 5 actions for sell from 1 to 5 shares, and 1 hold action. The following 3-action space with 10 shares is for buy, hold, and sell with 10 shares. The next 21-action space is for 10 actions for buy from 1 to 10 shares, 10 actions for sell from 1 to 10 shares, and 1 hold action. The two graphs on the top of Fig 11 show the entire performance of our experimental results with two reward functions on S&P500. The two graphs in the middle are on HSI, and last two graphs are for Eurostoxx50. These graphs show that multiple shares make more profit and the discrete action space model performs almost 29.3% better on an average than the three action space model with multiple shares in all of the cases. As a result, the result of the extended discrete action space is better than the case of multiple shares of a stock in the 3-action space.

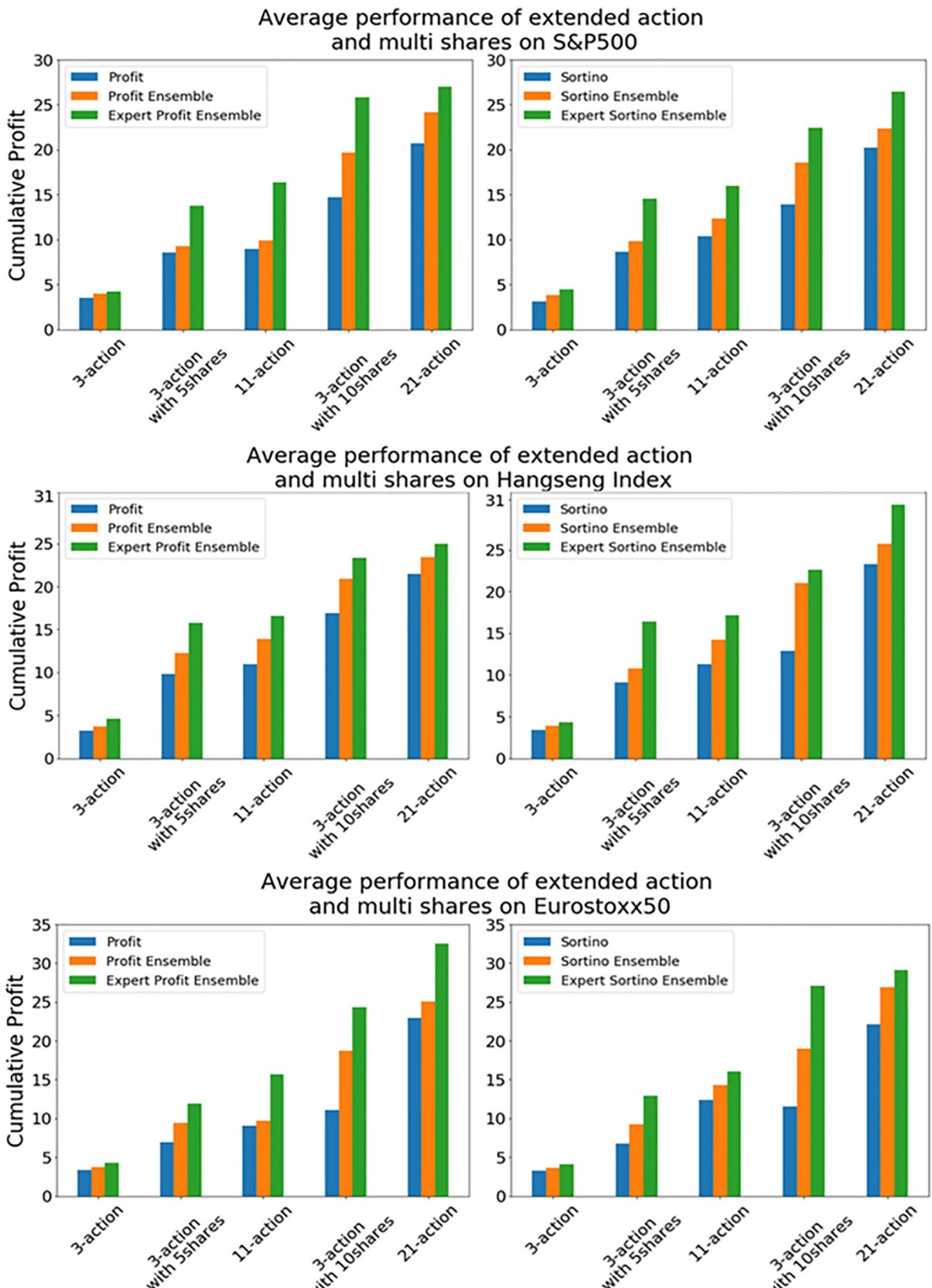

**Fig 11. Average performance of extended action and multi shares on three stock indices.**

Figs 12–14 indicate detail actions of the best result of the action-specialized expert ensemble model on each index in this study. We have only displayed the case of profit reward function because the case of Sortino reward function is similar. Each action space consists of two pictures. The first graph is the movement of each index during the actual test period and we mark each action on it. We can compare the actual movement trend with decisions of model. The second graph is the spread marking of actions to check a different number of actions. As seen in Figs 12–14, the actions decision of our proposed model closely-resembles the real price movement. We also can see the spread of actions, and it is evident that the network applies various actions according to the market situation and the extended discrete action space of the experiment. In detail analysis by each index, our model on S&P500 learns the upward trends and shows the result of continuously representing the buy action. The price movement of the other two indices is more volatile than S&P500, and these results show various action decisions depending on the strength of these signals.

### Analytic results of our whole experiments

Analyzing our results in connection with Table 4, results of Eurostoxx50's test period, which actually decreased by -16%, were generally higher profit than those of the other two indices. We think this is because the distribution characteristics of training data set and test set are similar. The kurtosis of two data sets of Eurostoxx50 is almost the same, and the gap of skewness is relatively small compared to S&P500 and HSI. For this reason, it seems to be able to learn relatively better than the other two index environments. In addition, S&P500 shows a relatively low volatility and upward trend during the train and test periods. This index pattern appears to be too simple to learn various patterns. As a result, in contrast to the environment of the other two indices that can learn a variety of information, the profit of trained model for S&P500 was lowest, even though the real index was highest in the same test period. Lastly, the HSI environment shows good results because the train and test data set movements are relatively enough to learn various patterns.

### Computational complexity

In the DRL approach, the computational complexity of the DRL model is important to understand the burden of the architecture. Thus, we analyze this in two ways; time & space complexity, and trade-off between training costs and performance.

### Time & space complexity

In the reinforcement learning, the time complexity is sublinear in the length of state period, and the space complexity is sublinear in the number of state space, action space, and steps per episode. These can be expressed as big $O$ notation, time complexity requires $O(n_T)$ space, where $n_T = n_e n_h$ is the total number of steps, $n_e$ is the number of episode, and $n_h$ is the number of steps per episode. Space complexity requires $O(n_s n_a n_h)$ space, where $n_s$ is the number of states and $n_a$ is the number of actions [50]. In addition, the computational complexity of DRQN can be calculated based on the complexity of the reinforcement learning and LSTM. The time and space complexity of an LSTM per time step is estimated as $O(n_w)$, where the $n_w$ is the number of weights of network [51]. Thus, the time complexity of DRQN is $O(n_w n_T)$ and the spatial complexity is $O(n_w n_s n_a n_h)$. Since the common ensemble method combines three single model of DQN, the time complexity is linear in the number of DQNs. However, it does not affect the space complexity of ensemble method. Therefore, the time complexity of ensemble method is estimated as $O(n_m n_T)$, where the $n_m$ is the number of base models, and the space complexity of ensemble method is estimated as $O(n_s n_a n_h)$, which is the same as the spatial

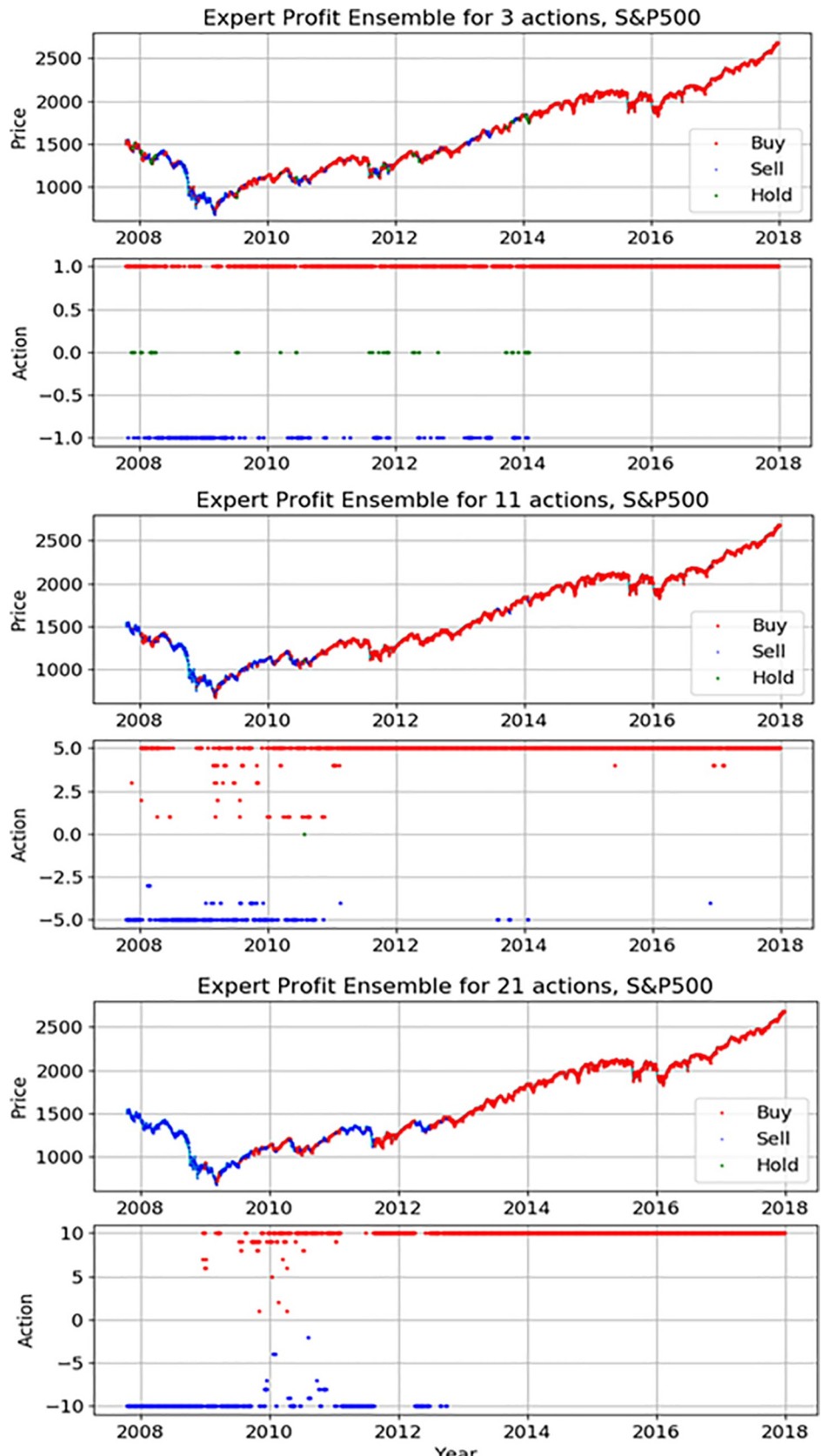

**Fig 12. The detail actions of the best expert ensemble models of each action on S&P500.**

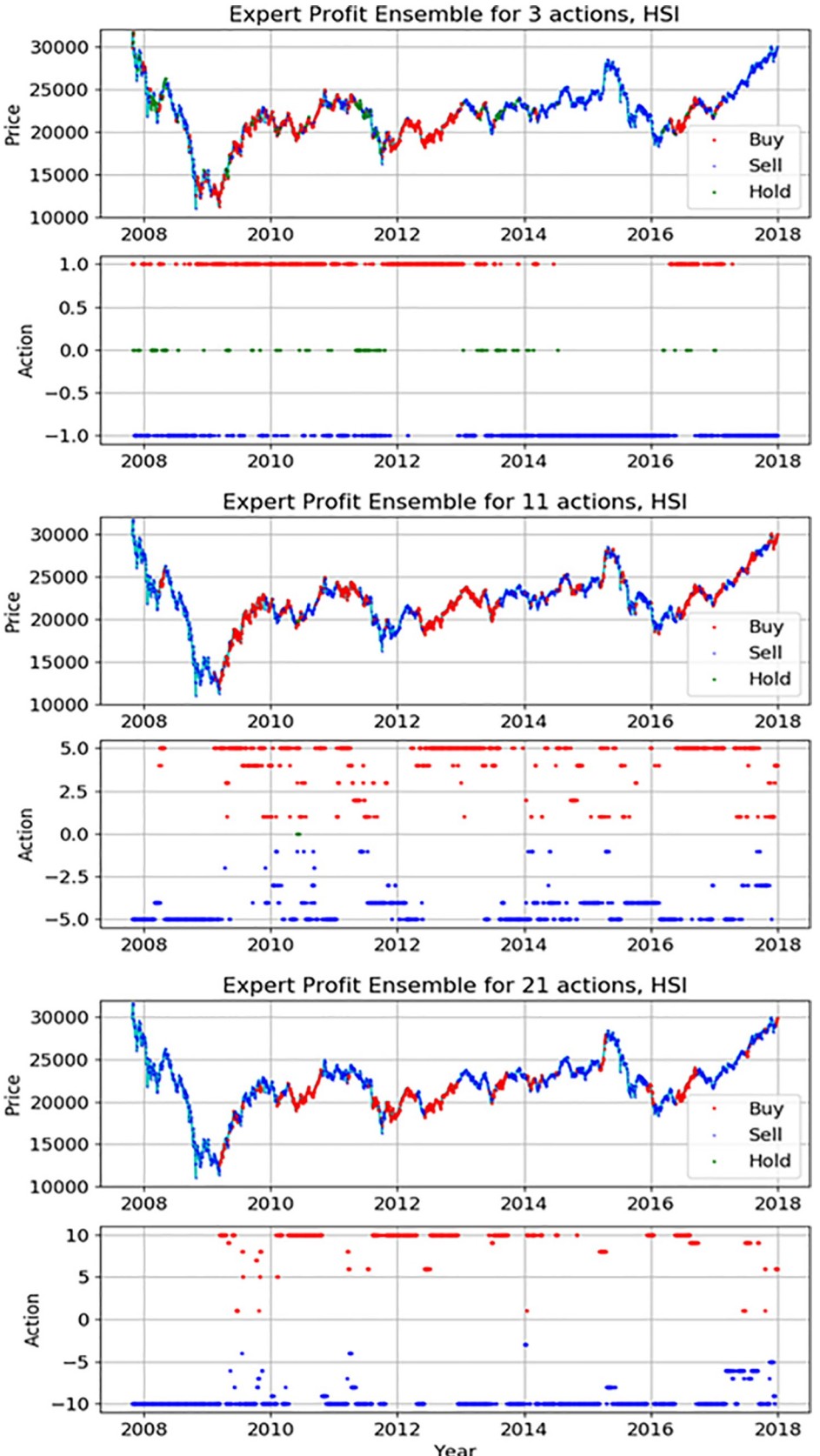

**Fig 13. The detail actions of the best expert ensemble models of each action on Hang Seng Index.**

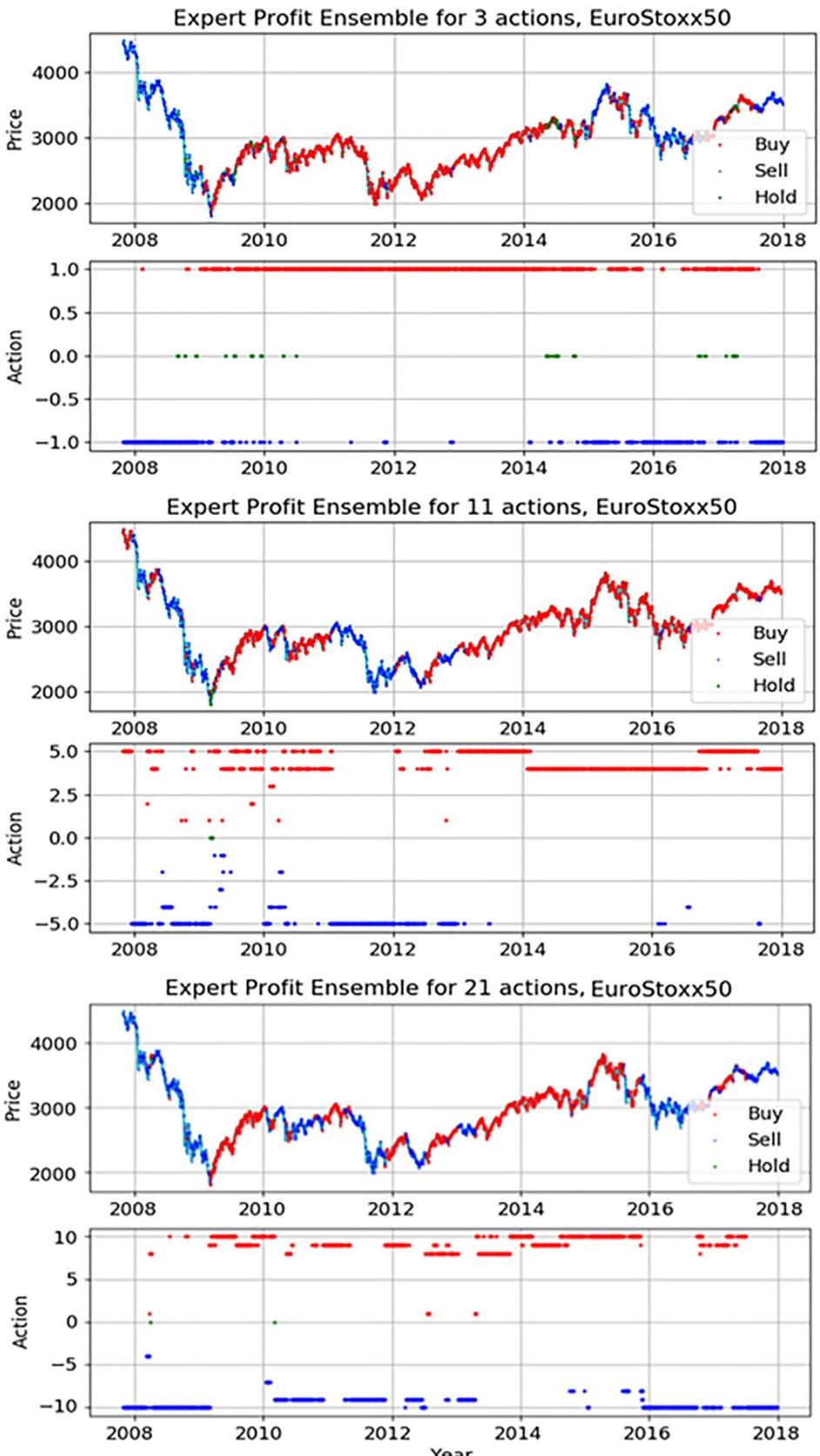

**Fig 14. The detail actions of the best expert ensemble models of each action on Eurostoxx50.**

**Table 12. The time and space complexity comparisons for our proposed algorithm and previous methods.**

| Method | Time complexity | Space complexity |
|---|---|---|
| DQN | $O(n_T)$ | $O(n_s n_a n_h)$ |
| DRQN | $O(n_w n_T)$ | $O(n_w n_s n_a n_h)$ |
| Common Ensemble | $O(n_m n_T)$ | $O(n_s n_a n_h)$ |
| Proposed method | $O(n_m n_T)$ | $O(n_s n_a n_h)$ |

complexity of DQN. Last, as the design of our proposed approach is the same as the common ensemble method, our proposed method has the same complexity of time and space as with the common ensemble method. We summarize the comparison of complexities in Table 12 below.

The inference time of the ensemble method takes almost 1.5ms longer than that of a single model in our experimental environment (Experimental Server Specifications: CPU: Xeon E5-2620, Ram: 64GB, GPU: GTX 1080 8-ways). Moreover, expert single models take almost 70s longer to learn than common models. The reason for the longer duration is a result of judging the range of profit and calculating reward values in the action-specialized expert model. Thus, in training, our proposed expert ensemble model takes about 3.5 times longer than a common single model and takes longer than the common ensemble model; however, its performance is better than the single and common ensemble model. When we tested our proposed models, since we focused on their performance, we did not train our proposed method simultaneously in an advanced parallel system. Thus, if we conduct our proposed method with parallel or distributed system, we can reduce the learning time of experiments better. The computational load is also a challenge to be solved in the reinforcement learning task. There are a number of studies on synchronous parallel systems, asynchronous parallel learning, and distributed reinforcement learning systems [52–56].

## Trade-off between training costs and performance

The trade-off between training costs and the performance were analyzed. First, we compared the performance of our proposed model with S&P500 by reducing the duration of training data to various lengths to discuss the trade-off between the different time period of training data set and performance. In more detail, the period of our training data set of the original experiment is 20 years (Jan 2, 1987–Dec 29, 2006) as seen in Table 3, however, we make 3 more training data sets which are different time periods of 5 years (Jan 2, 2002–Dec 29, 2006), 10 years (Jan 2, 1997–Dec 29, 2006), and 15 years (Jan 2, 1992–Dec 29, 2006) with same test data set period of 11-years. Fig 15(A)–15(C) show the performances of different training data sets and different actions, and Fig 15(D) displays the boxplot. In the boxplot, the red line is mean value and the green line is median. As seen in Fig 15, the longer training data set makes the better performance for all three discrete action spaces. In the more detailed explanation as seen in Fig 16(A), the performances of 10 years, 15 years, and 20 years are 1.6, 2.1, and 2.7 times better than the performance of 5 years.

In addition, we investigate the trade-off between training time cost and performance by measuring the training time of each different training data set and different action. We averaged the top five cumulative profits for each category and displayed it in Fig 16. The left scatter plot depicts the relative performance between the length of training data set. The training time and performance of five-year training data are based on 1, and results of remaining data sets are expressed as a ratio. The right scatter plot depicts the relative performance between different discrete action spaces. The training time and performance of the three-actions are based

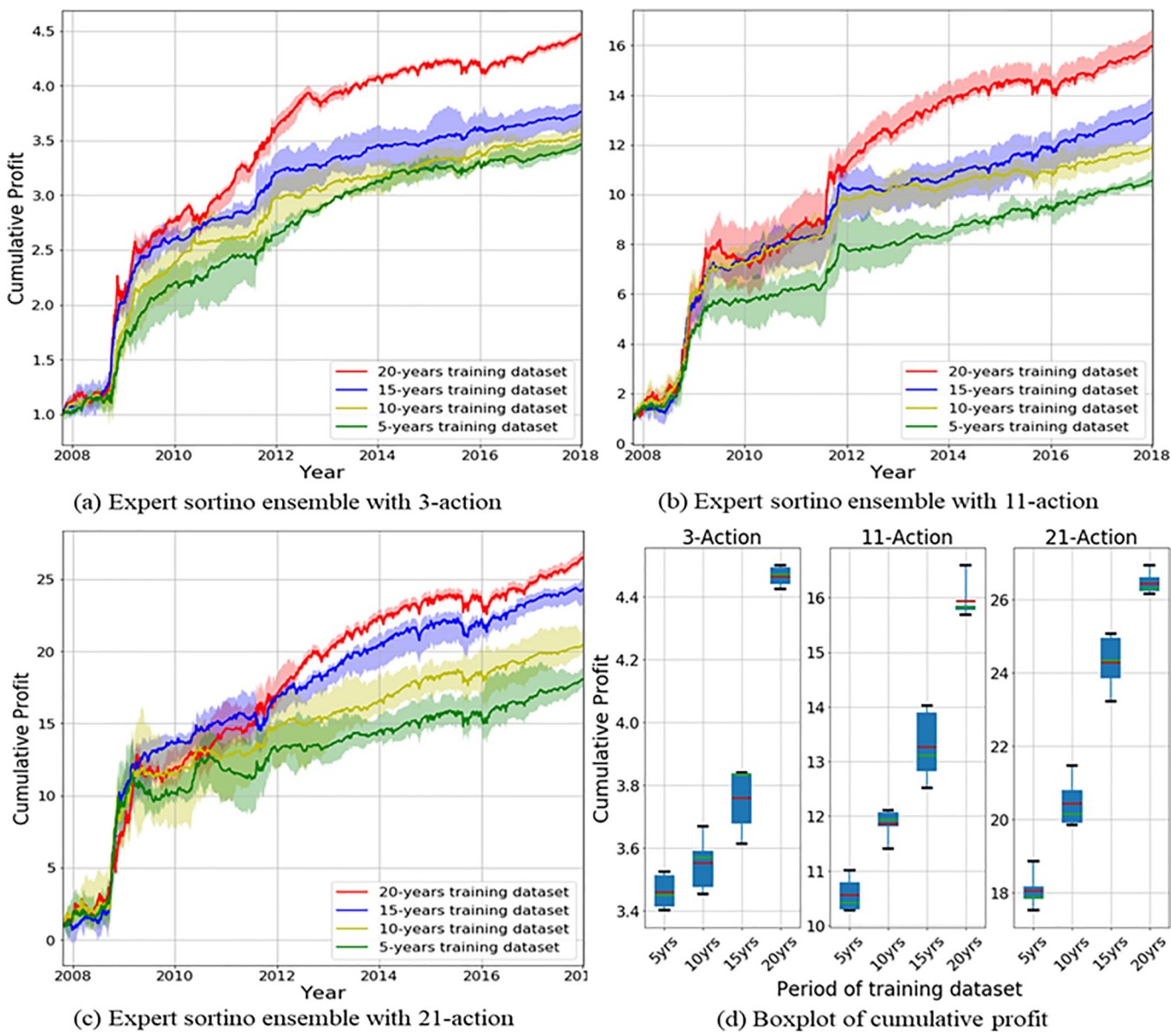

**Fig 15. Results of trade-off between different length of training period and performance with S&P500.**

on 1, and results of the remaining data sets are expressed as a ratio. The training times of 10 years, 15 years, and 20 years are 1.1, 1.23, and 1.42 times longer than the training times of five-years.

## Student's T-test of our proposed model on other methods

We experiment with the results of DQN, DRQN, and common ensemble of DQN to compare the results with previous studies and compare these three results with our action-specialized expert ensemble model. The experimental environment is only 3-action, and the data period is set to 20 years for training and 11 years for the test in three indices. Fig 17 shows the mean and standard error of five results from each model, and the performance of our method is excellent in all three indices. We conduct a student's T-test to see if this result is statistically significant.

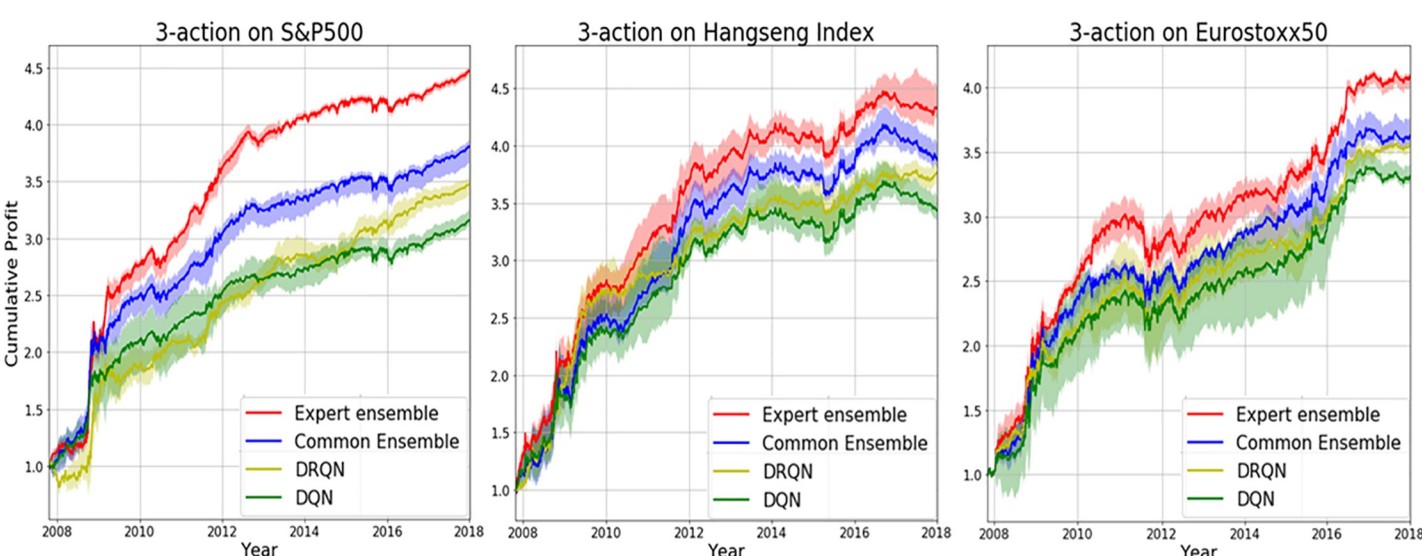

**Fig 16. Trade-off between training time costs and the performance.**

**Fig 17. Comparison of our proposed model's performance with other algorithms' performance on each index in 3-action space.**

**Table 13. 2-sample T-test of our proposed model on DQN, DRQN, common ensemble.**

| # of action | Index | S&P500 | | Hang Seng Index | | Eurostoxx50 | |
|---|---|---|---|---|---|---|---|
| | 2-samples | t | p-value | t | p-value | t | p-value |
| 3 | ESE / SE [57] | 11.42 | 7.8E−30 | 8.46 | 3.6E−17 | 8.68 | 5.3E−18 |
| | ESE / DQN [47] | 40.20 | 2.9E−301 | 18.44 | 2.6E−73 | 21.01 | 7.2E−94 |
| | ESE / DRQN [48] | 39.54 | 9.7E−296 | 20.57 | 3.2E−90 | 22.05 | 5.9E−103 |
| 11 | ESE / SE [57] | 27.48 | 2.1E−154 | 17.25 | 9.4E−65 | 14.42 | 3.3E−46 |
| | ESE / DQN [47] | 41.00 | 1.2E−302 | 38.94 | 1.8E−282 | 25.53 | 7.1E−135 |
| 21 | ESE / SE [57] | 10.92 | 2.0E−27 | 11.70 | 3.3E−31 | 5.65 | 1.7E−08 |
| | ESE / DQN [47] | 30.51 | 2.7E−185 | 16.44 | 4.9E−59 | 12.42 | 7.8E−35 |

ESE: expert Sortino ensemble, SE: Sortino ensemble (common ensemble)

Based on our model, we conducted the student's T-test (e.g., 2-sample T-test) on DQN, DRQN, and common ensemble models. As a result of the T-test in Table 13, all p-values are less than 0.05, so we can explain that our proposed model is statistically different from other models. Overall, we evinced the performance of our proposed model and the student's T-test and we could believe our experimental results.

## Conclusion

In this study, a new ensemble approach was proposed for automated trading systems using reinforcement learning—specifically, an action-specialized expert ensemble trading system—to improve performance. This ensemble model consists of action-specialized expert models specialized in buy, hold, and sell actions. Since we developed each specialized model individually, our proposed method can reflect investment behavior in each model differently and obtain various distribution effects. We verified our approach experimentally with three different stock indices: S&P500, HSI, and Eurostoxx50.

First, our proposed method displays better performance than the common ensemble and single models, and is 21.6% and 39.1% more effective than the common ensemble and the single models, respectively. Second, we compared the profits of our proposed model to common ensemble and single models to check the effect of the extension of the discrete action space. Briefly, results indicate an increase of 427.2% and 856.6% on the 11-action and 21-action models, respectively. Further, our extra experiments indicate that the extended action space is more efficient than multiple shares of a stock in the 3-action space. As the action space is extended, the training of each network becomes increasingly difficult. However, these results imply that our proposed method is well-trained with an extended discrete action space. Third, we analyzed the results of our proposed model with various reward functions: profit, Sharpe ratio, and Sortino ratio. As a result, the two ratios, which jointly consider profit and volatility, demonstrate a 9.6% better performance than the use of profit only in two-thirds of our experiments. We believe that both profit and volatility information is helpful in training the network.

In this study, we apply our proposed method to a trading system. Since our action-specialized expert model is developed based on actions with controlling reward function in DRL, it can be applied to other cases of DRL in other fields. For example, it is applicable to game fields. In the fighting game, it is possible to create expert models for an attack specialized expert model, a defense specialized expert model, and an evasion specialized expert model. Additionally, in the soccer video game, an ensemble model can be generated by making an attack specialized expert model and a defense specialized expert model. For more examples, because

autonomous driving must be realistic in many ways, action-specialized expert models can be created, such as expert models for recognizing moving vehicles, expert models for avoiding parking vehicles, expert models for driving well, and cornering or break expert models. In addition, in robot fields, we can develop an expert model for balancing, an expert model for walking, and an expert model for moving angles, and so on. Another application is to extend this study by first training the network with a discrete action space to a continuous action space using transfer learning. Therefore, we believe that it is possible to expand this study to various fields and further develop its application in financial fields in the future.

## Author Contributions

**Conceptualization:** Ha Young Kim.

**Data curation:** JoonBum Leem.

**Formal analysis:** JoonBum Leem, Ha Young Kim.

**Funding acquisition:** Ha Young Kim.

**Investigation:** JoonBum Leem, Ha Young Kim.

**Methodology:** JoonBum Leem, Ha Young Kim.

**Project administration:** Ha Young Kim.

**Resources:** Ha Young Kim.

**Software:** JoonBum Leem.

**Supervision:** Ha Young Kim.

**Visualization:** JoonBum Leem.

**Writing – original draft:** JoonBum Leem.

**Writing – review & editing:** Ha Young Kim.

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
