## [Decision Letter · Decision Letter 0]

10 Apr 2020

PONE-D-20-06706

Action-specialized expert ensemble trading system with extended discrete action space using deep reinforcement learning

PLOS ONE

Dear Dr. Kim,

Thank you for submitting your manuscript to PLOS ONE. After careful consideration, we feel that it has merit but does not fully meet PLOS ONE’s publication criteria as it currently stands. Therefore, we invite you to submit a revised version of the manuscript that addresses the points raised during the review process.

I recommend that it should be revised taking into account the changes requested by the reviewers. I would like to give you the last chance to revise your manuscript. To speed the review process, the manuscript will only be reviewed by the Academic Editor in the next round.

We would appreciate receiving your revised manuscript by May 25 2020 11:59PM. To enhance the reproducibility of your results, we recommend that if applicable you deposit your laboratory protocols in protocols.io, where a protocol can be assigned its own identifier (DOI) such that it can be cited independently in the future. For instructions see: http://journals.plos.org/plosone/s/submission-guidelines#loc-laboratory-protocols

We look forward to receiving your revised manuscript.

Kind regards,

Baogui Xin, Ph.D.

Academic Editor

PLOS ONE

Journal Requirements:

Reviewers' comments:

Reviewer's Responses to Questions

**Comments to the Author**

1. Is the manuscript technically sound, and do the data support the conclusions?

Reviewer #1: Yes

Reviewer #2: Yes

2. Has the statistical analysis been performed appropriately and rigorously? 

Reviewer #1: Yes

Reviewer #2: Yes

3. Have the authors made all data underlying the findings in their manuscript fully available?

Reviewer #1: Yes

Reviewer #2: Yes

4. Is the manuscript presented in an intelligible fashion and written in standard English?

Reviewer #1: Yes

Reviewer #2: Yes

5. Review Comments to the Author

Reviewer #1: This paper presents a new action-specialized expert ensemble method consisting of action specialized expert models for automated financial trading. Then, this approach was applied on three datasets. With respect to the obtained results, the new technique is able to provide acceptable results in terms of accuracies. In this reviewer’s opinion, the idea of using deep reinforcement learning with extended discrete action space looks promising, but several concerns should be clarified for possible publication. Some comments are listed below.

1. How about the time complexity and space complexity of the proposed algorithm when compared with the previous methods?

2. It is recommended to add some details about how to prevent overfitting.

3. Generally speaking, deep learning methods are time-consuming. Please add some considerations about the computational load of the proposed method.

Reviewer #2: The authors propose an expert ensemble method of 3-action models (buying, holding, and selling). Each model specialized for each action examined in the reinforcement learning for trading systems by learning different reward values under specific conditions. Then, the ensemble technique is used for marking the final action of the model. The proposed method is very interesting. However, there are some issues that the authors should clarify as follows.

1. From Fig. 2, the activation function of the proposed model should be given as well as the detail of ensemble action associated with the final decision of the system. Not many details of the proposed method are given, and they are scattering. The authors should elaborate the proposed method in one place, and the pseudo code or the flowchart of the algorithm should be given.

2. As the authors stated in page 5 (line #125) “When we are confident, we invest more, and when we are less confident, we adjust the quantity to take a relatively small risk and achieve a small loss or a small profit,” the authors should provide sufficient detail to support the statement.

3. The authors should provide more detail about the interrelation among the profit function (eq. 7), the predetermined positive constant (m), and the number of actions (3, 11, 21).

4. The definition of variables and parameters in the algorithm 1 should be given. They make it easy for the reader to follow the pseudo code.

6. PLOS authors have the option to publish the peer review history of their article (what does this mean?). If published, this will include your full peer review and any attached files.

Reviewer #1: No

Reviewer #2: Yes: Booncharoen Sirinaovakul

---

## [Author Response · Author response to Decision Letter 0]

26 Jun 2020

Detailed Response to Reviewers

Responses to reviewer #1:

We are grateful to reviewer 1 for the critical comments and useful suggestions that have helped us to improve our manuscript considerably. As indicated in the following responses, we have incorporated all these comments into the revised version of our manuscript.

Comment #1

This paper presents a new action-specialized expert ensemble method consisting of action specialized expert models for automated financial trading. Then, this approach was applied on three datasets. With respect to the obtained results, the new technique is able to provide acceptable results in terms of accuracies. In this reviewer’s opinion, the idea of using deep reinforcement learning with extended discrete action space looks promising, but several concerns should be clarified for possible publication. Some comments are listed below.

1. How about the time complexity and space complexity of the proposed algorithm when compared with the previous methods?

Response #1

We appreciate your helpful advice. We have calculated time complexity and space complexity of our proposed algorithm when compared with the previous methods, DQN, DRQN, and common ensemble method in Table 12. We have added the following sentences to the second paragraph of subsection Computational complexity of section Experimental results and discussion.

 In the reinforcement learning, the time complexity is sublinear in the length of state period, and the space complexity is sublinear in the number of state space, action space, and steps per episode. These can be expressed as big O notation, time complexity requires O(n_T) space, where n_T=n_e n_h is the total number of steps, n_e is the number of episode, and n_h is the number of steps per episode. Space complexity requires O(n_s n_a n_h) space, where n_s is the number of states and n_a is the number of actions [52]. In addition, the computational complexity of DRQN can be calculated based on the complexity of the reinforcement learning and LSTM. The time and space complexity of an LSTM per time step is estimated as O(n_w), where the n_w is the number of weights of network [53]. Thus, the time complexity of DRQN is O(n_w n_T) and the spatial complexity is O(n_w n_s n_a n_h). Since the common ensemble method combines three single model of DQN, the time complexity is linear in the number of DQNs. However, it does not affect the space complexity of ensemble method. Therefore, the time complexity of ensemble method is estimated as O(n_m n_T), where the n_m is the number of base models, and the space complexity of ensemble method is estimated as O(n_s n_a n_h), which is the same as the spatial complexity of DQN. Last, as the design of our proposed approach is the same as the common ensemble method, our proposed method has the same complexity of time and space as with the common ensemble method. We summarize the comparison of complexities in Table 12 below.

Table 12. The time and space complexity comparisons for our proposed algorithm and previous methods.

Method Time complexity Space complexity

DQN O(n_T) O(n_s n_a n_h)

DRQN O(n_w n_T) O(n_w n_s n_a n_h)

Common Ensemble O(n_m n_T) O(n_s n_a n_h)

Proposed method O(n_m n_T) O(n_s n_a n_h)

The citations for the response are as follows:

[52] Jin, C., Allen-Zhu, Z., Bubeck, S., & Jordan, M. I. (2018). Is q-learning provably efficient?. In Advances in Neural Information Processing Systems (pp. 4863-4873).

[53] Hochreiter, S., & Schmidhuber, J. (1997). Long short-term memory. Neural computation, 9(8), 1735-1780.

Comment #2

2. It is recommended to add some details about how to prevent overfitting.

Response #2

Thank you for helpful comment. We have explained further about how to prevent overfitting in this study and added the following sentences to the second paragraph of subsection Proposed ensemble model—action-specialized expert ensemble model of section Methodology.

 To prevent overfitting our proposed method and to train the network better, we used experience replay and epsilon-greedy in our DRL experiments. Regarding experience replay, all the experiences are saved in the replay memory in the shape of <s_t,a_t,r_t,s_(t+1)> during the training of the DQN network. Then, the replay memory is uniformly shuffled to make a mini-batch of random samples so that the mini-batch sample is not sequential. This eliminates the time dependency of subsequent training samples. In addition, the observed experience is reused to train when it is sampled repeatedly and improves data usage efficiency. Thus, it helps to avoid local minima and prevent overfitting. Next, the epsilon-greedy method is used to solve exploration exploitation dilemmas in DRL. The epsilon-greedy method chooses an action randomly with probability ε and the maximum Q-value action with probability (1-ε). The epsilon(ε) is decreased over an episode from 1 to 0.1. This will result in completely random moves to explore the state space maximally at the start of the training, which settles down to the fixed exploration rate of 0.1 at the end of the training. Therefore, the epsilon-greedy method helps to prevent overfitting or underfitting.

Comment #3

3. Generally speaking, deep learning methods are time-consuming. Please add some considerations about the computational load of the proposed method.

Response #3

We appreciate your helpful advice. We have tried to further explain the considerations of the computational load of the proposed method.

The deeper the network and the more the optimization, the better the performance of the model in deep learning. Thus, if we use more computational power and more time, we can obtain better results of learning. However, since the limit of computational power exists in general study, we should take these limitations into account. When we tested our proposed model, as we focused on their performance, we did not train our proposed method simultaneously in an advanced parallel system. Thus, if we conduct our proposed method with a parallel or distributed system, we can reduce the learning time of experiments better. The computational load is also a challenge to be solved in the reinforcement learning task and there are several studies about synchronous parallel systems, asynchronous parallel learning, and distributed reinforcement learning systems [1, 2, 3, 4, 5]. In addition, since our proposed model is based on an ensemble of 3 single models, we can apply an advanced parallel system to train 3 models simultaneously. Subsequently, the learning time of our proposed model can be reduced by approximately 3 times compared to an environment in which experiments cannot be performed in a parallel system.

We have modified and added some considerations about the computational load of the proposed method to the third paragraph of subsection Computational complexity of section Experimental results and discussion.

 The inference time of the ensemble method takes almost 1.5ms longer than that of a single model in our experimental environment (Experimental Server Specifications: CPU: Xeon E5-2620, Ram: 64GB, GPU: GTX 1080 8-ways). Moreover, expert single models take almost 70s longer to learn than common models. The reason for the longer duration is a result of judging the range of profit and calculating reward values in the action-specialized expert model. Thus, in training, our proposed expert ensemble model takes about 3.5 times longer than a common single model and takes longer than the common ensemble model; however, its performance is better than the single and common ensemble model. When we tested our proposed models, since we focused on their performance, we did not train our proposed method simultaneously in an advanced parallel system. Thus, if we conduct our proposed method with parallel or distributed system, we can reduce the learning time of experiments better. The computational load is also a challenge to be solved in the reinforcement learning task. There are a number of studies on synchronous parallel systems, asynchronous parallel learning, and distributed reinforcement learning systems [54–58].

The citations for the response are as follows:

1) Nair, A., Srinivasan, P., Blackwell, S., Alcicek, C., Fearon, R., De Maria, A., ... & Legg, S. (2015). Massively parallel methods for deep reinforcement learning. arXiv preprint arXiv:1507.04296.

2) Kretchmar, R. M. (2002). Parallel reinforcement learning. In The 6th World Conference on Systemics, Cybernetics, and Informatics.

3) Mnih, V., Badia, A. P., Mirza, M., Graves, A., Lillicrap, T., Harley, T., ... & Kavukcuoglu, K. (2016, June). Asynchronous methods for deep reinforcement learning. In International conference on machine learning (pp. 1928-1937).

4) Liu, T., Tian, B., Ai, Y., Li, L., Cao, D., & Wang, F. Y. (2018). Parallel reinforcement learning: a framework and case study. IEEE/CAA Journal of Automatica Sinica, 5(4), 827-835.

5) Lauer, M., & Riedmiller, M. (2000). An algorithm for distributed reinforcement learning in cooperative multi-agent systems. In In Proceedings of the Seventeenth International Conference on Machine Learning.

Responses to reviewer #2:

We are grateful to reviewer 2 for the critical comments and useful suggestions that have helped us to improve our manuscript considerably. As indicated in the following responses, we have incorporated all these comments into the revised version of our manuscript.

Comment #1

The authors propose an expert ensemble method of 3-action models (buying, holding, and selling). Each model specialized for each action examined in the reinforcement learning for trading systems by learning different reward values under specific conditions. Then, the ensemble technique is used for marking the final action of the model. The proposed method is very interesting. However, there are some issues that the authors should clarify as follows.

1. From Fig. 2, the activation function of the proposed model should be given as well as the detail of ensemble action associated with the final decision of the system. Not many details of the proposed method are given, and they are scattering. The authors should elaborate the proposed method in one place, and the pseudo code or the flowchart of the algorithm should be given.

Response #1

Thank you for your valuable comment. We have modified and added further explanation in the revised manuscript. We have collated the information and described our proposed method in section Methodology. We have integrated and revised the second paragraph of subsection Proposed single model—action-specialized expert model of section Methodology and the first paragraph of section 5.2 to the second paragraph in subsection Proposed single model—action-specialized expert model.

 The reward function of the expert model is expressed by the following Eq (10).

r_t^expert={█(r_t×m,&〖if a〗_t is an expert action in the range of profit @according to Table 4 in Section 5 below@r_t,&otherwise)┤ (10)

where r_t is the reward value, and m is the predetermined positive constant and m≥1. We will explain this constant m in Section 5. For the m (predetermined positive constant), we constructed the range of profit based on profit distribution, and divided it into buy, hold, and sell actions based on the threshold. We set the threshold at 0.3% because it is used as a general transaction cost, and it is possible to prevent a loss by choosing a holding strategy if it does not generate more than 0.3% profit. In Eq (10), m is applied step-by-step—depending on the importance of profit and frequency. As frequency varies according to the profit interval, the absolute value of profit is important. Table 2 indicates the design of predetermined positive constants of the expert model by profit interval. Because of Due to this conditional reward function r_t^expert, we can control the reward, and through this equation, we used the adjusted reward value to develop the proposed single expert model.

We have moved Table 2 from section 5.2 to section subsection Proposed single model—action-specialized expert model of section Methodology and added the following sentences to the third paragraph of subsection Proposed single model—action-specialized expert model of section Methodology.

Table 2. Predetermined positive constants of the expert model by profit interval.

Expert model : Buy Expert model : Hold Expert model : Sell

Range of profit Predetermined positive 

constant (m) Range of profit Predetermined positive 

constant (m) Range of profit Predetermined positive constant (m)

-∞ – 0.3% 1 -∞ – -0.3% 1 -∞ – -5% 10

0.3 – 1% 3 -0.3 – 0.3% 7 -5 – -3% 7

1 – 2% 5 0.3 – ∞% 1 -3 – -2% 6

2 – 3% 6 -2 – -1% 5

3 – 5% 7 -1 – -0.3% 3

5 – ∞% 10 -0.3 – ∞% 1

 By controlling the reward value with m, we can create the enhanced model for specific action according to the reward value. In detail, we modify the reward function by multiplying it and m for learning the action-specialized expert model when the model makes a correct decision. For example, according to Table 2, the buy-specialized expert model obtains the enhanced reward value that is m times larger than the common reward value when its decision is correct in the range of profit. The enhanced compensation is only applied when the decision is correct. If the decision is wrong, the reward value is small or under zero but not at the enhanced penalty value. In other words, the model obtains larger reward values when it works well in the specific action, and so becomes the specific action-specialized expert model. Thus, each action-specialized expert model of buy, hold, and sell can be created by controlling the enhanced reward function with m. In addition, we apply the extended discrete action space and it makes the reward value larger than the 3-action space. The extended action space helps the model determine whether the action is strong or weak. Specifically, the buy action in the 3-action model is only one, whereas, the buy actions in the 11-action model are five—which means buying 1 to 5 shares. The action of buying 1 share is similar to a weak buy action whereas the action of buying 5 shares indicates a strong buy action. In addition, since the reward function of the action-specialized expert model with extended action space is defined as multiplying reward value, m and extended action (the number of shares), the action-specialized expert model can obtain more various reward values, which have a wide range. Thus, if the model can detect the degree of obvious patterns, which can be the direction and magnitude of dynamic market movements from input state, then it can determine how many shares to buy or sell of a stock depending on the detected degree by choosing the correct extended action.

And we have added the following sentences to the end of first paragraph of subsection Proposed ensemble model—action-specialized expert ensemble model of section Methodology.

 Figure 2 indicates the process of common ensemble model and our proposed model. The reward of common model is the raw value of profit or Sortino ratio and common ensemble consists of these models. The reward of our proposed model, on the other hand, is controlled by an additional value which is compensation for the expert action under specific condition. In this way, it consists of three different action-specialized expert models based on reinforcement learning. In Figure 2, the colored boxes represent enhanced expert action of each expert single model. In the common ensemble method, performance substantially improves because an ensemble of a plurality of networks can be averaged to reduce the deviation of the resulting network. Unlike the common ensemble method that combines similar models, our proposed ensemble method combines buy-, hold-, and sell-specialized single expert models to improve performance. For instance, our proposed ensemble model functions similarly to three experts from different fields cooperatively making decisions with unifying opinions. Thus, each expert model yields a different inference or decision with the same input; however, our ensemble method improves performance. When we employ it, we use the soft voting ensemble method, which can avoid loss of information [16]. The soft voting method equation is as follows.

〖Output〗_tj=softmax(a_t [j])=(exp⁡(a_t [j]))/(∑_(j=1)^J▒〖exp⁡(a_t [j])〗) (11)

where 〖Output〗_tj is converted from the softmax function with a_t [j] at time t. a_t is the action as outputs of the model and J is the number of outputs. The softmax function normalizes each action value, which is the Q-value in DQN in the expert model, between 0 and 1. Since the sum of all the action values after applying the softmax function becomes 1, each value of output layer of DQN in the expert model indicates the probability of each action. For the final decision of the expert ensemble model, the Q-value of DQN in each expert model takes the softmax function. After that, the average of the activated Q-values of buy, hold, and sell-specialized expert models become the final outputs of the expert ensemble model, that is, the Q-values of the expert ensemble model. Thus, the action of the highest Q-value of the expert ensemble model is selected as the final decision. To describe our proposed method in detail, the DQN algorithm for our model is provided in Algorithm 1 below.

We have moved Algorithm 1 from section 5.3 to section subsection Proposed ensemble model—action-specialized expert ensemble model of section Methodology and added the following sentences to the second paragraph of subsection Proposed ensemble model—action-specialized expert ensemble model of section Methodology.

 To describe our proposed method in detail, the deep Q network algorithm for our model is provided in Algorithm 1 below.

Algorithm 1 Deep Q Network for Single Models and Action Specialized Expert Models

Hyperparameters: M - size of experience replay memory, R - repeat step of training, X_train - training data set, T_train - episode of training data, m - predetermined positive constant, mini-batch size - 64, C - episode of updating target Q ^ network, T_test - episode of test data.

Parameters: w - weights of Q network, w^- - weights of target Q ^ network

Variables: Total Profit - cumulative profit as performance measure, s_t - state space at time t, a_t - action space at time t, r_t - reward at time t, r_t^expert - reward for expert model at time t

Initialize replay memory M 

Initialize the Q network with random weights w

Initialize the target Q ^ network with weights w^-=w

Training Phase

for STEP = 1,R do

Set training data set X_train

 Total profit=1

 for episode = 1,T_train do

 Set state s_t

Choose action a_t following ε-greedy strategy in Q 

 If common single model == True then

 r_t= r_t

 Else if action specialized expert model == True then

If range of profit == True and expert action == True then

 r_t^expert= r_t×m (Equation (10))

Else if range of profit == True and expert action == False then

 r_t^expert= r_t

Else if range of profit == False and expert action == True then

 r_t^expert= r_t

Else if range of profit == False and expert action == False then

 r_t^expert= r_t

 end if

end if

Set next state s_(t+1)

 If len(M) == max_memory then

 Remove the oldest memory from M

 Else

 Store memory (s_t,a_t,r_t,s_(t+1)) in replay memory buffer M

 end if

 for each mini-batch sample from buffer M do

 Q(s_t,a_t ) □(←) Q(s_t,a_t )+α∙(r_(t+1)+γ max┬a⁡Q ^ (s_(t+1),a)-Q(s_t,a_t ))

 Total profit ← 〖Total profit+profit〗_t. 

 end for

 in every C episodes, reset Q ^=Q, i.e., set weights w^-=w

end for

end for

Clear replay memory buffer M

Test Phase

Set test data set X_test for Online learning

Set Total profit=1

for episode = 1, T_test do

Repeat Training Phase with R=1

end for

Ensemble Phase

Prepare three models of each expert action model

Ensemble these models by soft voting at each time t (Inference time ensemble)

 To prevent overfitting our proposed method and to train the network better, we used experience replay and epsilon-greedy in our DRL experiments. Regarding experience replay, all the experiences are saved in the replay memory in the shape of <s_t,a_t,r_t,s_(t+1)> during the training of the DQN network. Then, the replay memory is uniformly shuffled to make a mini-batch of random samples so that the mini-batch sample is not sequential. This eliminates the time dependency of subsequent training samples. In addition, the observed experience is reused to train when it is sampled repeatedly and improves data usage efficiency. Thus, it helps to avoid local minima and prevent overfitting. Next, the epsilon-greedy method is used to solve exploration exploitation dilemmas in DRL. The epsilon-greedy method chooses an action randomly with probability ε and the maximum Q-value action with probability (1-ε). The epsilon(ε) is decreased over an episode from 1 to 0.1. This will result in completely random moves to explore the state space maximally at the start of the training, which settles down to the fixed exploration rate of 0.1 at the end of the training. Therefore, the epsilon-greedy method helps to prevent overfitting or underfitting.

Comment #2

2. As the authors stated in page 5 (line #125) “When we are confident, we invest more, and when we are less confident, we adjust the quantity to take a relatively small risk and achieve a small loss or a small profit,” the authors should provide sufficient detail to support the statement.

Response #2

Thank you for your valuable comment. We have modified and added further explanation with supportive statements to the fourth and fifth paragraph of section Introduction.

 To verify our proposed method and check its robustness, we included more action spaces by discretizing, which determines the number of multiple shares of a stock to buy or sell by itself. Previous studies have either only studied the three actions or proceeded to a continuous action space [1–7,15]. It is well known that as the output of the model network increases, learning becomes more difficult [17]. In a previous study, however, discretizing action spaces yielded better performance than applying continuous action spaces [18]. Thus, in this study, we have extended the number of actions from 3 to 11 and 21, and the quantity of actions has been is increased by 5 and 10, respectively. Moreover, we expect the network to be able to recognize market risk and control the quantity by itself. One of the purposes of our research is to create a more profitable automated trading system that allows for more investment when data-driven patterns are clearer, such as real investors investing more boldly as compared to the information they get from the market. Therefore, our model is designed to learn various patterns from data and vary actions to improve performance increase profit according to signal strength the magnitude of reward value we designed. Compared to the existing 3-action models, existing models could not represent the diversity of actions depending on signal strength of results obtained the reward value of the model trained from the data. For example, we could not determine whether the buy signal in the 3-action model is strong or weak. In contrast, our proposed system with more discrete actions is much more profitable than the 3-action system because it can buy or sell more, depending on the market situation. More specifically, the trading model with 21 actions can ideally increase profits by up to 10 times, since it can trade more quantities (up to 10 times) for stronger signals than weaker ones. In view of the financial aspect, more risks must be taken to obtain higher profits. On this basis, we wondered what would happen if the amount of buying and selling was increased to an action space in a trading system with reinforcement learning. If we increase the maximum number of shares when we have a model with three actions, that is, buy, hold, and sell, we can obtain the maximum expected profit from the model. This can be seen as a leverage effect. However, it is not so simple in the financial market because if we are confident of the accuracy of this model, we can leverage it up to the maximum quantity. On the other hand, if the model is wrong, we will correspondingly lose as much leverage. Therefore, we designed the reinforcement learning model to implement a trading system that self-adjusts leverage within the maximum quantity by considering risk as well as profit by expanding the action space by the maximum quantities, which are 5 and 10. As a result, we have produced many experimental results that can support this. When we are confident, we invest more, and when we are less confident, we adjust the quantity to take a relatively small risk and achieve a small loss or a small profit. 

 If the extended action space model can capture the level of obvious patterns from the dynamic market data, it can decide the quantities of investment by itself—depending on the captured level of information. As we give the adaptive signal to our model through controlling reward values by extending action space, we expect that our model can analyze more detailed market information, which includes the degree of both direction and magnitude of market movements. If the proposed model is confident in the market condition, it will invest more in the market. Whereas, if the model is less confident in the market condition, it will adjust the quantity to take a relatively small risk in order to achieve a small loss or a small profit. As a result In this regard, we have produced many experimental results that can support this. Many reinforcement learning-based trading system studies surveyed have three action spaces, and our research is meaningful as the pioneering study that attempts this. As expected, our results indicate that Deep Reinforcement Learning (DRL) can learn not only three actions, but also various other actions, depending on the strength of the network signals.

Comment #3

3. The authors should provide more detail about the interrelation among the profit function (eq. 7), the predetermined positive constant (m), and the number of actions (3, 11, 21).

Response #3

Thank you for your valuable comment. We have provided more details about the interrelation among the reward function (eq. 7-9), the predetermined positive constant (m), and the number of actions (3, 11, 21). We have added the following sentences to the third paragraph of subsection Proposed single model—action-specialized expert model of section Methodology.

 By controlling the reward value with m, we can create the enhanced model for specific action according to the reward value. In detail, we modify the reward function by multiplying it and m for learning the action-specialized expert model when the model makes a correct decision. For example, according to Table 2, the buy-specialized expert model obtains the enhanced reward value that is m times larger than the common reward value when its decision is correct in the range of profit. The enhanced compensation is only applied when the decision is correct. If the decision is wrong, the reward value is small or under zero but not at the enhanced penalty value. In other words, the model obtains larger reward values when it works well in the specific action, and so becomes the specific action-specialized expert model. Thus, each action-specialized expert model of buy, hold, and sell can be created by controlling the enhanced reward function with m. In addition, we apply the extended discrete action space and it makes the reward value larger than the 3-action space. The extended action space helps the model determine whether the action is strong or weak. Specifically, the buy action in the 3-action model is only one, whereas, the buy actions in the 11-action model are five—which means buying 1 to 5 shares. The action of buying 1 share is similar to a weak buy action whereas the action of buying 5 shares indicates a strong buy action. In addition, since the reward function of the action-specialized expert model with extended action space is defined as multiplying reward value, m and extended action (the number of shares), the action-specialized expert model can obtain more various reward values, which have a wide range. Thus, if the model can detect the degree of obvious patterns, which can be the direction and magnitude of dynamic market movements from input state, then it can determine how many shares to buy or sell of a stock depending on the detected degree by choosing the correct extended action.

Comment #4

4. The definition of variables and parameters in the algorithm 1 should be given. They make it easy for the reader to follow the pseudo code.

Response #4

Thank you for your suggestion. We have added the definitions of hyperparameters, parameters, and variables in the Algorithm 1 and modified the following Algorithm 1 in the subsection Proposed single model—action-specialized expert model of section Methodology. 

Algorithm 1 Deep Q Network for Single Models and Action Specialized Expert Models

Hyperparameters: M - size of experience replay memory, R - repeat step of training, X_train - training data set, T_train - episode of training data, m - predetermined positive constant, mini-batch size - 64, C - episode of updating target Q ^ network, T_test - episode of test data.

Parameters: w - weights of Q network, w^- - weights of target Q ^ network

Variables: Total Profit - cumulative profit as performance measure, s_t - state space at time t, a_t - action space at time t, r_t - reward at time t, r_t^expert - reward for expert model at time t

Initialize replay memory M 

Initialize the Q network with random weights w

Initialize the target Q ^ network with weights w^-=w

Training Phase

for STEP = 1,R do

Set training data set X_train

 Total profit=1

 for episode = 1,T_train do

 Set state s_t

Choose action a_t following ε-greedy strategy in Q 

 If common single model == True then

 r_t= r_t

 Else if action specialized expert model == True then

If range of profit == True and expert action == True then

 r_t^expert= r_t×m (Equation (10))

Else if range of profit == True and expert action == False then

 r_t^expert= r_t

Else if range of profit == False and expert action == True then

 r_t^expert= r_t

Else if range of profit == False and expert action == False then

 r_t^expert= r_t

 end if

end if

Set next state s_(t+1)

 If len(M) == max_memory then

 Remove the oldest memory from M

 Else

 Store memory (s_t,a_t,r_t,s_(t+1)) in replay memory buffer M

 end if

 for each mini-batch sample from buffer M do

 Q(s_t,a_t ) □(←) Q(s_t,a_t )+α∙(r_(t+1)+γ max┬a⁡Q ^ (s_(t+1),a)-Q(s_t,a_t ))

 Total profit ← 〖Total profit+profit〗_t. 

 end for

 in every C episodes, reset Q ^=Q, i.e., set weights w^-=w

end for

end for

Clear replay memory buffer M

Test Phase

Set test data set X_test for Online learning

Set Total profit=1

for episode = 1, T_test do

Repeat Training Phase with R=1

end for

Ensemble Phase

Prepare three models of each expert action model

Ensemble these models by soft voting at each time t (Inference time ensemble)

---

## [Editor Report · Decision Letter 1]

1 Jul 2020

Action-specialized expert ensemble trading system with extended discrete action space using deep reinforcement learning

PONE-D-20-06706R1

Dear Dr. Kim,

We’re pleased to inform you that your manuscript has been judged scientifically suitable for publication and will be formally accepted for publication once it meets all outstanding technical requirements.

Kind regards,

Baogui Xin, Ph.D.

Academic Editor

PLOS ONE
---

## [Editor Report · Acceptance letter]

8 Jul 2020

PONE-D-20-06706R1 

Action-specialized expert ensemble trading system with extended discrete action space using deep reinforcement learning 

Dear Dr. Kim:

I'm pleased to inform you that your manuscript has been deemed suitable for publication in PLOS ONE. Congratulations! Your manuscript is now with our production department. 

Kind regards, 

on behalf of

Professor Baogui Xin 

Academic Editor

PLOS ONE